# Glycosaminoglycans, Instructive Biomolecules That Regulate Cellular Activity and Synaptic Neuronal Control of Specific Tissue Functional Properties

**DOI:** 10.3390/ijms26062554

**Published:** 2025-03-12

**Authors:** James Melrose

**Affiliations:** 1Graduate School of Biomedical Engineering, University of New South Wales, Sydney, NSW 2052, Australia; james.melrose@sydney.edu.au; 2Raymond Purves Bone and Joint Research Laboratories, Kolling Institute of Medical Research, Northern Sydney Local Health District, Royal North Shore Hospital, St. Leonards, NSW 2065, Australia; 3Sydney Medical School, Northern, University of Sydney at Royal North Shore Hospital, St. Leonards, NSW 2065, Australia

**Keywords:** glycosaminoglycans, human disease, basement membrane, dystroglycan, neurophysiology, synaptic stabilization, neural networks, heparan sulfate, keratan sulfate, chondroitin sulfate

## Abstract

Glycosaminoglycans (GAGs) are a diverse family of ancient biomolecules that evolved over millennia as key components in the glycocalyx that surrounds all cells. GAGs have molecular recognition and cell instructive properties when attached to cell surface and extracellular matrix (ECM) proteoglycans (PGs), which act as effector molecules that regulate cellular behavior. The perception of mechanical cues which arise from perturbations in the ECM microenvironment allow the cell to undertake appropriate biosynthetic responses to maintain ECM composition and tissue function. ECM PGs substituted with GAGs provide structural support to weight-bearing tissues and an ability to withstand shear forces in some tissue contexts. This review outlines the structural complexity of GAGs and the diverse functional properties they convey to cellular and ECM PGs. PGs have important roles in cartilaginous weight-bearing tissues and fibrocartilages subject to tension and high shear forces and also have important roles in vascular and neural tissues. Specific PGs have roles in synaptic stabilization and convey specificity and plasticity in the regulation of neurophysiological responses in the CNS/PNS that control tissue function. A better understanding of GAG instructional roles over cellular behavior may be insightful for the development of GAG-based biotherapeutics designed to treat tissue dysfunction in disease processes and in novel tissue repair strategies following trauma. GAGs have a significant level of sophistication over the control of cellular behavior in many tissue contexts, which needs to be fully deciphered in order to achieve a useful therapeutic product. GAG biotherapeutics offers exciting opportunities in the modern glycomics arena.

## 1. Introduction

Cells exist in an environment where they receive signals from extracellular matrix (ECM) components filtered through the glycocalyx or directly from adjacent interconnected cells, which have instructive properties over cellular behavior [1,2,3,4]. The ability to perceive such signals stems from glycosaminoglycans (GAGs) attached to sensory proteins on the cell surface, which allow the cell to perceive perturbations in cellular microenvironments [2], and this facilitates the orchestration of responsive cellular biosynthetic alterations in ECM components to replenish matrix components depleted by disease, degradative, or traumatic events [5]. This feed-back mechanism allows for the coordination of anabolic biosynthetic events during tissue morphogenesis and in tissue repair responses in remodeling ECMs following traumatic tissue damage and also the regulation of essential physiological processes, which regulate tissue homeostasis [6].

The importance of the ECM for cellular protection and tissue function in dense weight-bearing, tension-resisting and high-shear tissues is clearly evident and stresses the important functional roles ECM PGs play in these tissues [7,8]. Instructional cues from the ECM are important in cellular mechanosensing; cell–ECM communication allows cells to sense their microenvironment and to respond to any ECM perturbations in health and disease [9,10,11]. This allows cells to regulate tissue homeostasis and maintain optimal tissue functional properties [5], mediated by their component GAGs and reflected in the use of GAGs in biomedicine. However, our knowledge of their full therapeutic value is incomplete [12]. Dystroglycan (DG), a highly glycosylated glycoprotein, provides an interconnection between the ECM and cytoskeleton, which facilitates bioresponsive cell–ECM communication [13,14]. DG-HS-PG (neurexins, pikachurin, Eyes-shut) interactions mediated by laminin-G PG modules in their core proteins provide synaptic stabilization, synaptic plasticity and specificity of synaptic interactions through interactions with a vast array of synaptic adhesion effector proteins [15,16]. PGs and GAGs also have specific roles to play in vascular disease [17] and in the normal functioning of the CNS/PNS following injury [18,19,20,21]. The biodiversity of GAGs points to their varied functional properties in specific tissues in health and disease and in wound repair [22]. Cell surface PGs such as the syndecan and glypican families [23] have important roles to play in interactions with growth factors and morphogens [24,25] and control inflammation [26,27], cellular proliferation and differentiation and cell signalling in tissue morphogenesis, skeletogenesis and in wound repair processes [28]. The syndecans and glypicans act as growth factor receptors and signalling platforms [29,30] and are regulators of cellular behaviour [31], with important roles in tissue regeneration [32] and roles in angiogenesis and endothelial cell biology [30,33].

### 1.1. GAGs Convey Structural and Functional Properties to Tissues

GAGs in mammals are *O*-linked to serine or threonine residues or *N*-linked through asparagine residues to PG core proteins (Figure 1). Some mucin protein backbones can also act as acceptor molecules for the addition of GlcNAc and D-Gal residues, and selective sulfotransferases can sulfate these, leading to the attachment of KS to some classes of mucins. These have been shown to have sensory properties in some tissues [34]. Elasmobranch fish species (sharks, rays) contain sensory pits (Ampullae of Lorenzini) located on regions of the skin filled with an ultrasensitive sensory mucin-like KS gel [35], which allows for the detection of the electric fields preyfish species emit through muscular exertion [36]. This process is known as electrolocation and allows for sensitive detection of preyfish species, even in turbid poor-visibility conditions [37,38]. Some fish species also use electrolocation with low-intensity electric signals as a mechanism for communication between other members in their group; this provides information on sex, status within the group hierarchy and sexual maturity of individuals when searching for a mate [39,40]. Electric eels can generate high-intensity electric fields sufficient to immobilize prey species as a hunting mechanism [41]. Terrestrial animals have lost the ability to electrolocate, except for two monotreme species, the duck-billed platypus and echidna in Australia. These have mechano- and electroreceptors in their bill structures, which they use in electrolocation and mechano-sensing [42,43]. The platypus is a nocturnal feeder, which feeds with its eyes closed, and electrolocation is essential for its food gathering activities.

### 1.2. GAG Biodiversity

Five classes of GAGs have been identified in mammals on the basis of their generic repeat disaccharide structures and assembly to PG core protein GAG acceptor groups (Figure 1). Some of the glycans in repeat disaccharides are sulfated at specific positions, providing very diverse interactive properties to GAGs (Figure 1a–f). Hyaluronan (HA) is the only GAG which is unsulfated and is unattached to a PG core protein. The sulfation along a single CS, HS or KS GAG chain is not uniform, and specific domains within GAG chains act as important functional determinants, facilitating interaction with growth factors, receptors, morphogens and structural ECM components. Gene mutations in the transporter proteins and sulfate metabolic enzymes involved in GAG biosynthesis cause a number of skeletal dysplasias due to disruption in the normal PG sulfation patterns and the functional properties GAGs provide to tissues, demonstrating the importance of these sulfation motifs in tissue functional properties [44]. These dysplasias include achondrogenesis type 1B, atelosteogenesis type 2, dystrophic dysplasia, recessive multiple epiphyseal dysplasia, brachyolmia type 1 and 4, spondoepimetaphyseal dysplasia, spondyloepiphyseal dysplasia with congenital dislocations, Ehlers-Danlos syndrome (musculocontractural type 1), osteochondrodysplasia, brachydactyly with overlapping malformed digits and neurofascioskeletal syndrome with or without renal agenesis [45,46].

### 1.3. KS Biodiversity

Keratan sulfate (KS) chains contain non-sulfated polylactosamine, monosulfated and disulfated regions; however, the size of each of these regions can vary, leading to considerable size and charge heterogeneity in the KS structure (Figure 1b–e) [47]. Sulfation in KS is predominantly on GlcNAc residues in monosulfated regions, while in disulfated regions, the D-galactose residues are also sulfated. D-Gal sulfotransferase only acts on a KS disaccharide if the adjacent GlcNAc is first sulfated, giving rise to a disulfated disaccharide; thus, heterogeneous distributions of mono- or disulphation or non-sulphation regions can occur in a KS chain. GlcNAc undergoes sulfation more frequently than Gal in the KS disaccharide. Like all GAGs, the sulfation status of KS defines its functional properties. The KS linkage regions to PG also vary as *O*-glycans linked to serine or threonine residues or as *N*-glycan residues attached to asparaginee. KSI chains are end-capped with neuraminic acid, disulfated GalNAc and Gal disaccharides sulfated at C6 [48,49] and are also internally modified with L-Fucose residues to a variable degree. KS is the only branched GAG. In porcine corneal KS, the C-6 branch of the linkage oligosaccharide is extended, but the C-3 branch is prematurely truncated and terminated in a single lactosamine region capped by sialic acid [50]. Two non-sulfated lactosamine disaccharides are present nearest the reducing terminus of the C-6 branch, but 10–12 sulfated GlcNAc disaccharides are found on the more distal part of the chain. Collectively, this information has allowed for the classification of KS chains into three types: (i) KSI (corneal KS) *N*-linked to asparagine residues, (ii) KSII *O*-linked to serine or threonine residues, found in articular and other weight-bearing cartilages, and has been sub-classified into type IIA and IIB in weight-bearing and non-weight-bearing cartilages [47], (iii) KSIII in brain tissue is also *O*-linked to serine or threonine residues, and this linkage region involves mannose residues.

KSIII in brain tissues is the second richest source of KS after the cornea [47,51]. KSI was the first form of KS identified, and the cornea is the richest tissue source of this GAG, which was named corneal KS [52]; however, this form of KS also decorates a number of PGs with a widespread tissue distribution in tissues other than the cornea; thus, its naming is a historical misnomer. KSI is present as N-linked KS chains in fibromodulin, lumican, PRELP (prolargin), keratocan and osteoadherin [47,50,53,54]. KSIIA is found in articular cartilage, meniscus and intervertebral disc; type IIB KS is found in laryngeal, tracheal, nasal cartilages, and low-weight-bearing cartilages [55,56]. With ageing, there is an overall increase in the sulfation of KS chains, and while CS chains in aggrecan are reduced in size with ageing, KS chains are increased in length [57]. The significance of these changes in the GAG chain structure on aggrecan biology is not known. A further fucosylated form of KS has also been detected in the CS1 and CS2 regions of the aggrecan core protein using MAb 3D12/H7 [58]; however, its specific roles are not known.

Monoclonal antibodies to KS (Table 1) react with extracts from most mammalian tissues, with at least sixteen ECM PGs substituted with KS, and several cell-associated KS-PGs have been identified [59]. The lack of uronic acid in KS and variable sulfation results in charge heterogeneity in KS [47,60]. A number of related poly-*N*-acetyl lactosamine-modified proteins exist; however, these are not sulfated [61]. The development of MAb R10G, 1B4 and 294-1B1 allows for the detection of KS-PG species of low sulfation and mucin-like proteins containing lactosamine regions containing GlcNAc and Gal residues that are sulfated. Formerly, antibodies such as 5D4 and MZ-15, which detect high charge density KS glycoforms, were routinely used in this research area; however, these do not detect such low-sulfation isoforms of KS. Thus, a new aspect of the biology of low-sulfation KS-PGs is now emerging with the development of these newer KS antibodies [34,37,62]. KS expression is elevated in many tumors, including pancreatic and lung cancer [63,64]. Highly sulfated KS is also highly expressed in astrocytic tumors and in glioblastoma [65,66]. KS in cancer has been shown to be highly associated with advanced tumor grade and poor prognosis [63,64]. Higher-level expression of KS in primary pancreatic tumors and in lung metastatic deposits is associated with worse patient survival. Epithelial mucin (MUC1) [67] and an isoform of CD44 [68] contain KS chains and KSPGs, which are widely distributed in tissues and have a diverse range of functions [47].

**Table 1 ijms-26-02554-t001:** Antibodies developed to KS epitopes, illustrating their diversity in structure.

Antibody Clone	Antibody Specificity	Reference
5-D-4	Hexa sulfated KS octa-saccharide and a linear dodecasaccharide containing N-sulfated glucosamine in KS multisulfated regions	[69,70]
MZ-15	Hepta and octa-saccharide KS oligosaccharides in multisulfated KS regions	[70]
IB-4	Tetrasulfated hexasaccharide in linear KS mono-sulfated region	[70]
R10G	Low sulfation KS in mono-sulfated regions	[71,72,73]
294-1B1	Low sulfation KS decorating podocalyxcin	[74,75]
3D12/H7	Sulfated fucosylated poly-N-acetyllactosamine linkage region epitope distributed throughout the CS1 and CS2 region of cartilage aggrecan	[58]
6D2/B5	Fucosyl-KS epitope	[76]
SV2	High sulfation KS chains on SV2 PG	[77,78]
EFG-11	Tri KS disaccharides	[79]
1/14/16H9	Specific equine KS antibody	[80,81]
BKS-1 (+)	KS neo-epitope, 6-sulfated GlcNAc adjacent to a nonsulfated lactosamine disaccharide in reducing terminal PG linkage region exposed by keratanase-1 pre-digestion.	[82]
TRA-1-60	Epitope is sensitive to neuraminidase, keratanase-I/II, and endo-β-D-galactosidase. Epitope identified Galβ1-3GlcNAcβ1-3Galβ1-4GlcNAc and Galβ1-3GlcNAcβ1-3Galβ1-4GlcNAcβ1-6(Galβ1-3GlcNAcβ1-3)Galβ1-4Glc this oligosaccharide, is expressed on podocalyxcin on pluripotent embryonic stem cells	[83,84,85,86,87]
60-mG_2a_-f	A cancer-specific anti-podocalyxin monoclonal antibody and effective anti-cancer therapeutic.	[88]
TRA-1-81	Epitope is resistant to neuraminidase but sensitive to endo-β-D-galactosidase and keratanase-I/II. Epitope is terminal Galβ1-3GlcNAcβ1-3Galβ1-4GlcNAc and Galβ1-3GlcNAcβ1-3Galβ1-4GlcNAcβ1-6(Galβ1-3GlcNAcβ1-3)Galβ1-4Glc oligosaccharides on cell surface podocalyxcin of pluripotent embryonic stem cells	[83,84,85,86,87]
SSEA-1	Cell surface glycan of murine embryonic pluripotent stem cells, expressed on PG and glycoprotein core proteins and bioactive lipids	[89]
“i” antigen	Human autoantibody to a non-branched epitope in non-sulfated poly-*N*-acetyllactosamine	[90,91,92,93]
“I” antigen	Human autoantibody to a branched epitope in non-sulfated poly-*N*-acetyllactosamine regions of KS	[90,91,92,93]
4C4	Highly sulfated KS on embryonic tumor cell podocalyxcin	[94]

**Figure 1 ijms-26-02554-f001:**
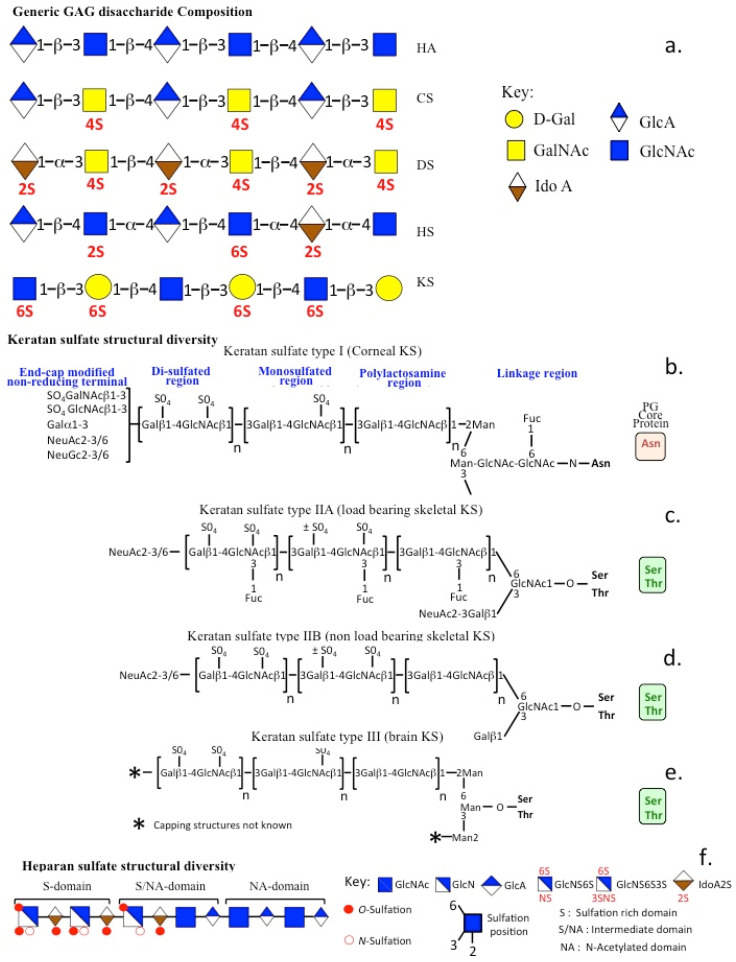
GAG biodiversity and the generic repeat disaccharides (**a**). KS types I (**b**), IIA (**c**), IIB (**d**) and III (**e**) and the linkage GAG acceptor regions upon which GAG chains are assembled by a range of sulfo- and glycosyl transferases. HS displays variable structures in the sulfated (S-domain), sulfated/N-acetylated mixed domain (S/NA domain) and N-acetylated domain (NA-domain) (**f**). The variable sulfation patterns in KS and HS are prominent features and responsible for the large repertoire of ligands interactive with these GAGs. Standard symbol nomenclature for glycans (SFNG) icons is used to depict specific glycan sub-structures [95,96]. KS and HS chain heterogeneity regulates physiological processes, tissue function and tissue morphogenesis. Figure modified from [63] with permission.

### 1.4. KS Chain Heterogeneity in a Range of PGs and Mucins

KS chains in PGs display size and charge heterogeneity and range in size from 3 to 20 kDa, although larger KS chains > 60 kDa have also been reported for neuronal synaptic vesicle PG (SV2) [78]. High- and low-sulfation isoforms of KS have also been identified. SLRP family members (PRELP, osteoadherin, chondroadherin, epiphycan, and osteoglycin) have small low-sulfation KS chains (3–4 disaccharides, 3–5 kDa) [47,59]. PRELP from the cornea and sclera is present as 60–116 and 55–60 kDa proteins. Digestion with endo-β-D-galactosidase converts corneal PRELP to 45–50 kDa in size and scleral PRELP to a molecular weight of 50 kDa [97]. Digestion with *N*-glycanase further reduced these to 42–45 and 45KDa sized core proteins, demonstrating the presence of *N*-linked KS chains of low sulfation.

KS from the KS-rich region of aggrecan is 10–20 kDa in size and is *O*-glycosylated, while KS chains in the G1, G2, and interglobular domain (IGD) are small (3–4 disaccharides, 3–5 kDa) and may be *N*- or *O*-linked to the core protein. They are also of lower sulfation than KS in the KS-rich region [98,99]. The KS chains of normal corneal KSPGs are ~15 kDa in size and of low sulfation and have roles in tissue morphogenesis [100]. Mucin glycoproteins and PRELP have lactosamine residues, which can be sulfated on C6 of GlcNAc or Gal, leading to the formation of small KS chains of 3–4 disaccharides in size. Low sulfation impedes KS chain elongation. Mucosal tissues of the gastrointestinal, respiratory, reproductive, urinary tracts, and the surface of the eye have an enormous surface area and act as a selective molecular barrier at the epithelial surface, which acts as a barrier to the exterior environment and infection [101].

Glycanated mucins have been suggested as a therapeutic target in cancer therapy [102,103,104]. KS has been found as a component of the tumor environment [63]. Levels of mucins substituted with sulfated glycans are elevated in ovarian epithelial tumors [105]. Low-sulfation KS expression is elevated in human pancreatic [64] and ovarian cancer [74,105], metastatic lung tumors [106], head and neck squamous cell carcinoma [107], bladder cancer [108]. Podocalyxcin in testicular cancer has been shown to contain a low-sulfation KS isoform [75]. The KS chains of SV2 PG are significantly larger than the aforementioned KS chains, ranging in size from 27 to 140 kDa. This may contribute to the storage properties of SV2 for neurotransmitters [78]. KS is an electrosensory GAG in neural tissues [109] and in mucin glycopolymer gels [34,37] used in electrolocation in elasmobranch fish species as well as the duck-billed platypus and echidna [37]. KS-PGs modulate neuronal migration and have instructional properties in axonal migration and the formation of correctly assembled functional neural networks [20,21,59,110].

### 1.5. Low-Sulfation KS-PGs and Glycoconjugates

Ampullae of Lorenzini actin microfilament electrosensory mucoid KS glycoconjugate gel [36] found in sharks and rays and phosphacan/RPTP-ζ [111], Lumican/Keratocan, Mucins, PRELP, mimecan, podocalyxcin and tectorin associated with sensory hair cells in the cochlea [112] all contain low-sulfation motif KSPGs. Mucins in the reproductive tract also contain high- and low-sulfated KS, and these may have neuronal signaling roles [34]. A large 220 kDa KS PG present in human cervical mucous may have a role in the reorganization of the cervical ECM during the reproductive cycle. In the cervical mucous of early or nonpregnant women, it is associated with cervical ripening and quantitative and qualitative changes in the endometrium [113]. Endometrial MUC1 carries sulfated lactosaminoglycan chains that are hormonally regulated and show increasing abundance in the secretory phase [114]. The 5D4 KS epitope is abundant at the luminal epithelial cell surface until the implantation phase, when it disappears, first from patches of cells and then all together.

MUC1 carries sulfated lactosaminoglycans, identifying the luminal epithelial compartment as a site of unique MUC1 glycosylation and independent regulation [67]. Glycosylation and the negative charge associated with sialo- and sulfoglycans may be important in the regulation of embryo attachment. OVCAR-3 epithelial cells isolated from the malignant ascites of a patient with progressive ovarian adenocarcinoma also do not express highly sulfated KS but express low-sulfated glycans containing tandem GlcNAc-6-O-sulfated LacNAc units [74]. Podocalyxcin in human testicular embryonic carcinoma also expresses low-sulfated KS chains [75]. In the cochlea, tectorin a cochlear KSPG associated with the tectorial membrane and sensory hair cells does not react with mab 5D4 to a highly sulfated KS of corneal and skeletal muscle PG [112]. mab 5D4 selectively stains the upper surface of the tectorial membrane, Hensen’s stripe and the mucus layer overlying the respiratory epithelium. MMM and LMM tectorins may be unique to the cochlea. HMM may be a low charge density KS PG antigenically related to either the mucins or a more specific component of the olfactory mucus layer [112]. The sulfation status of KS chains may, thus, have consequences in electrosensory tissues and in neuronal regulation [21,37,47,59,63,109,115].

## 2. Diversity of KS-PG Functional Roles in Health and Disease

### 2.1. Aggrecan

Aggrecan (ACAN) [57] is the most abundant extracellular CS-KS PG in cartilage, representing ~35% of its dry weight, and it plays a key role in its biophysical and biomechanical properties [116,117,118,119,120,121]. ACAN forms aggregates with HA that are important in tissue hydration and in the generation of internal hydrostatic pressure and viscoelastic hydrodynamic properties, which allows tissues to withstand compressive loading. In the ECM of cartilage, aggrecan–HA complexes are constrained within a type II collagen meshwork, and this collagen network is inflated by solvated aggrecan–HA complexes. The collagen network also contributes to the biophysical properties of cartilage. The water-imbibing properties of aggrecan–HA are a function of the high fixed charge density they carry from the localization of sulfated GAG and the carboxyl groups of the uronic acid polyelectrolytes, described by the Gibbs–Donnan effect [122]. The Gibbs–Donnan effect describes the unequal distribution of permeant charged ions on either side of a semipermeable membrane, which occurs in the presence of impermeant charged ions. The high negative charge density provided by GAG localization in cartilage provides this tissue with hydrophilic properties, high osmotic swelling pressure and conformational flexibility, which collectively function to absorb fluctuations in biomechanical stresses experienced by cartilage during variable movement of an articular joint and the dissipation of biomechanical forces. Aggrecan is targeted by a number of MMPs and ADAMTS 4 and 5, which degrade cartilage aggrecan in OA/RA, and a significant reduction in biomechanical competence leads to impaired joint articulation [123,124]. In the brain, aggrecan provides space-filling properties, hydration and contributes to the formation of perineuronal nets, which regulate synaptic plasticity and cognitive processes [125,126]. Aggrecan–HA macro-aggregates are important for the hydration and compartmentalization of brain tissues, providing niche environments conducive to the optimal functioning of brain cell populations, and are neuroprotective [127]. Aggrecan plays an important role in the organization of the neural ECM binding and organizing HA to the cell surface through interactions with link protein and tenascins, forming a large quaternary aggregated complex that has cell instructive properties in embryonic brain development [127]. Other members of the lectican CS-PG family (versican, neurocan, brevican) also form aggregates with HA; however, the solvation volume of these complexes is not as extensive as aggrecan–HA complexes. Aggrecan is also present in cardiac jelly, developing heart valves, and blood vessels during cardiovascular development. Aggrecan contributes to the biophysical and regulatory properties of the cardiovascular ECM [128].

### 2.2. SV2

SV2 occurs as a 100 and 250 kDa 12-span transmembrane PG, containing three large KS chains with unique transport and neurotransmitter storage roles as a smart gel within neural synaptic vesicles. Three variably distributed SV2 isoforms, (SV2A, SV2B, and SV2C) bind the neurotransmitters glutamate, GABA, dopamine, choline, and the Ca^2+^ sensor protein synaptotagmin [129,130] and store these within synaptic vesicles [131]. The interaction of SV2 with synaptotagmin mediates Ca-dependent neurotransmitter release from synaptic vesicles [132,133]. The abnormal regulation of neurotransmitter release occurs in epilepsy, Schizophrenia, Alzheimer’s, and Parkinson’s disease [130]. SV2-KS interactions with Ca^2+^ regulate synaptic functions [133]. SV2 is phosphorylated on serine and threonine residues by serine/threonine kinases [134], which affects cytoskeletal organization, cell signaling and neurotransmitter interactions. The KS chains of SV2 interact with neurotransmitters, and the fucosylated proteins synapsin and synaptophysin tether synaptic vesicles and aid in their coordinated transport and subsequent release of neurotransmitters into the synaptic gap [131,135]. The depolymerization of the synaptic gap cell membrane facilitates fusion of the synaptic vesicles and release of their contents into the synaptic gap, where they are taken up by receptors on communicating neurons in the network [136]. SV2 is a synaptic vesicle neurotransmitter transporter and storage PG; it occurs as H and L forms. SV2A, SV2B, SV2C paralogs share 60% sequence and 80% structural homology. SV2A is expressed in peripheral sympathetic synapses, where it controls transmitter release. SV2B is the primary paralog expressed in the retina. SV2s are expressed in motor axons that innervate muscle fibers; SV2B and SV2C are expressed in all motoneurons [78,137,138,139,140,141,142].

### 2.3. Phosphacan

Phosphacan occurs as three CS, CS-KS PG isoforms, which may also contain the HNK-1 trisaccharide. Phosphacan modulates cell interactions and developmental processes in nervous tissue through binding to cell surface and ECM proteins [143]. Phosphacan/DSD-1 is the mouse homolog of phosphacan and is a developmentally regulated glial PG splice variant of RPTP-ζ [144,145,146]. Phosphacan modulates axonal extension during neuritogenesis [147]. Phosphacan is the soluble ectodomain of RPTP-ζ, existing as three splice variants [143,148]: (i) a full-length form containing N-terminal carbonic anhydrase-like domain, fibronectin type III repeat domain, CS attachment region and two intracellular phosphatase domains; (ii) a soluble form lacking the intracellular tyrosine phosphatase domains; and (iii) a truncated form lacking the majority of the CS attachment region. Phosphacan is widely expressed in the CNS in the cerebrum, hippocampus, cerebellum, spinal cord, olfactory system, and retina. The monoclonal antibody, Cat-315, detects a HNK-1 epitope on RPTPζ, in the developing brain [149]. RPTPζ and phosphacan regulate key developmental neural processes, including proliferation [150], differentiation [151], cell adhesion and migration [152], axonal guidance and neurite outgrowth [147,153], myelination [154,155] and participate in cognitive functional processes [109]. The extracellular domains of RPTPζ/phosphacan bind a wide array of ligands important for normal CNS development, including pleiotrophin [156], midkine [157], tenascin [158], NCAM, Ng-CAM and contactin [159,160]. Phosphacan has instructive roles in axonal guidance and neurite extension [161].

### 2.4. Podocalyxcin

Podocalyxcin is an anti-adhesive transmembrane polysialylated KS PG, with essential roles to play in neural development [162,163], and is a marker of human embryonic and induced pluripotent stem cells [164]. It is upregulated in glioblastoma formation and in astrocytomas [65,66,165,166,167,168,169] and has been developed as a prognostic factor for various cancers [170,171]. The sulfation status of the KS chains on podocalyxcin on normal embryonic cells and tumor cells differs, with the former expressing a low-sulfation KS detected by MAb R-10G [71,72,73], while tumor cells produce a high-sulfation KS chain [166], detected by antibodies such as 5-D-4, MZ-14 or 4C4 [69,70,94]. Podocalyxcin co-localizes with synapsin and synaptophysin in synapse vesicle formations [162]. Synaptophysin is a major synaptic vesicle protein, which coordinates the endocytosis of synaptic vesicles during neural stimulation [172]. Synapsin tethers synaptic vesicles to cytoskeletal components, preventing premature vesicle release into the synaptic gap coordinating neurotransmitter release from the synaptic vesicles [173,174,175,176].

### 2.5. Lumican and Fibromodulin

The LRRs of FMOD and LUM interact with an extensive range of ECM proteins that organize and stabilize tissues. FMOD and LUM reciprocally regulate collagen fibrillogenesis but do not share functional equivalence [177]. FMOD promotes the formation of thick collagen fibers, providing mechanical strength to tissues such as the sclera, whereas LUM forms thin regularly spaced collagen fibers, providing optical clarity in the cornea [178,179]. N-terminal sulfated tyrosine residues in LUM interact with FGF-2, TSP-1, MMP-13, NC4 domains of collagen IX and IL-10, binding to collagen and promoting fibril formation [179,180]. FMOD sequesters TGF-β, controlling its bio-availability [181], binds C1q and activates the complement system [182]. LUM impedes tumor growth through its MMP inhibitory activity, affects focal adhesions and the migration and growth of tumor cells through interaction with α2β1 integrin and inhibitory effects on angiogenesis [183,184,185].

### 2.6. Keratocan

KERA, keratocan, originally isolated from cornea, is a 50 kDa SLRP that has been immunolocalized in IVD and the spinal cord [82] and is a 37 kDa PG in other tissues. Keratocan contains low-sulfation KS-I side chains and has neuroregulatory and matrix organizational roles during spinal development, mediated by growth factor and morphogen interactions.

### 2.7. Prolargin/PRELP

PRELP (proline-arginine-rich end leucine-rich repeat protein) shares 36% identity with FMOD and 33% with LUM, and it anchors perlecan and type I collagen to basement membranes, and type II collagen to cartilage [186,187]. PRELP has an N-terminal Arg and Pro extension, and its alternative name is prolargin. PRELP can contain low-sulfation KS chains in some tissue contexts. PRELP is an anchoring component in many basement membranes and binds type I and II collagens and perlecan to stabilize the basement membrane [188]. PRELP has also been identified in brain tumor biopsies [189] and in gene expression and microarray studies, which distinguish glioblastoma and meningioma cases [190]. PRELP and biglycan are deposited precisely at myofibers surrounded and/or invaded by inflammatory cells in sporadic inclusion body myositis and in polymyositis [191]. PRELP has also been identified in the anterior pituitary gland and is apparently produced by pericytes in this tissue [192].

### 2.8. Chondroadherin

The C-terminal domain of CHAD is a regulator of osteoclast motility. Its α2β1 integrin binding sequence inhibits pre-osteoclast migration, decreasing osteoclastogenesis and bone resorption, but has no effect on osteoblasts. CHAD is a novel regulator of bone metabolism that may find applications in the treatment of osteoporosis [193,194]

### 2.9. Osteomodulin

Silencing OMD gene expression significantly suppresses alkaline phosphatase activity, mineralized nodule formation and osteogenesis-associated gene transcription. OMD acts as a positive coordinator of osteogenesis through BMP2 and SMAD signaling. WNT1 is an osteo-anabolic factor and transcriptionally activates the expression of OMD, which regulates type I collagen fibril formation in vitro. OMD is located in bone tissue and has important roles in bone mineralization. OMD reduces the diameter and changes the shape of collagen fibrils, regulating the ECM during bone formation [195,196,197,198].

### 2.10. Osteoglycin

Studies linking osteoglycin to insulin resistance, bone development, cardiovascular disease and pancreatic cancer suggest it may be a novel marker of muscle, pancreatic, and bone metabolism. Furthermore, osteoglycin may have roles in the regulation of energy homeostasis and additional roles in metabolic disorders [199,200]. Knowledge of osteoglycin is incomplete, and further research is required to confirm these possibilities.

## 3. HS Biodiversity

HS is an ancient molecule that evolved through strict natural selection criteria as a highly interactive molecule in the glycocalyx [201]. The immense molecular diversity of HS equips it with interactive capability [202,203], molecular recognition, information storage and signal transfer properties [201], capable of modulating cellular behavior and regulating essential physiological life processes [204,205,206] and is reflected in a range of antibodies that have been raised to HS (Table 2). The persistence of HS from the earliest days of evolution testifies the essential roles HS played in life processes [207,208]. It should be noted that the biosynthesis of HS requires 20+ biosynthetic enzymes [209]; this is a major investment by cells in a significant level of genetic information. The biosynthetic pathways catalyzed by these enzymes also have significant energetic demands; thus, the longevity of HS through a protracted period of evolution is significant [210]. The molecular diversity of HS is a property of the glyco-code contained in its GAG chains, which facilitates highly specific interactions with a broad range of functional soluble and structural proteins [210,211,212]. After heparin, HS is the most heterogeneous GAG, and it displays a significant level of structural complexity [213]. HS is assembled from a number of non-sulfated, mono-sulfated, di-sulfated, tri-sulfated and tetra-sulfated hexa glucuronic acid-glucosamine disaccharides, which are acetylated and then de-acetylated to variable degrees in different regions of the HS chain. HS is not synthesized by a template-driven process, but its diversity reflects the spatiotemporal expression of HS biosynthetic enzymes in tissues. These are variably expressed in tissues; their expression is controlled by the Hippo cell signaling pathway and TAZ/YAP transcription factors during tissue development and repair responses following trauma in health and disease [11,214,215,216,217,218,219].

Table 2 provides information on some antibodies that have been raised to HS. These illustrate some of its structural complexity. The binding of HS by phage display antibodies involves interactions between negatively charged carboxylic acid, *N*- and *O*-sulfate groups and regions on the protein surface that contain positively charged amino acid side chains, such as lysine and arginine. The binding of phage antibodies to HS is complex. Multiple HS structures bind the basic surfaces on the variable regions of the antibodies [220]. Phage display antibodies recognize conformationally defined HS epitopes [221], rather than the sequence alone, and are sensitive to changes in both charge distribution and conformation following cation binding [222,223]. Phage display antibodies are widely used to follow HS expression in tissues and cells [223,224,225,226,227]. Cations alter phage display antibody binding profiles to HS mediated by changes in HS chain conformation, demonstrated by circular dichroism spectroscopy [228,229,230]. Native HS structures, expressed on the cell surfaces of neuroblastoma and fibroblast cells, exhibit altered antibody binding profiles following exposure to low mM concentrations of cations [222].

**Table 2 ijms-26-02554-t002:** HS antibodies.

Antibody Clone	Tissue Antibody and Epitope Reactivities	Refs.
HepSS1	Predominantly localised to tissues rich in basement membrane.	[231]
JM13	Predominantly localised to tissues rich in basement membrane. JM13 binding epitope requires the presence of 2-O-sulfated glucuronic acid residues	[232]
JM403	Immunolocalised to basement membrane and some cell surface epitopes, in bovine lung, aorta and Human aorta. JM403 binding epitope is critically dependent on N-unsubstituted GlcN residues,	[233]
10E4	Immunolocalised to basement membrane and some cell surface epitope in human aorta, bovine intestine and kidney. The 10E4 epitope requires N-unsubstituted glucosamine residues. 10E4 binds to native “mixed” HS domains containing both N-acetylated and N-sulfated disaccharide units Ab reactivity is destroyed by heparinase III digestion	[234]
3G10	HS neoepitope generated by heparinase III ldigestion of HS chains. The 3G10 desaturated uronate stub epitope is attached to a HS disaccharide unit attached to the reducing terminal HS linkage region to core protein.	[234]
MAb865	N-acetylated regions in HS	[235,236]
JM72	HS-PG core protein epitope	[232,237]
Phage display antibodies	Antibody tissue immunoreactivity	
HS4C3	HSMC3 shows strong localization in bovine intestine and kidney, *O*-and *N*- linked HS epitopes are important for Ab binding	[221]
HS4D10	HS4D10 epitope immunoreactivity is strong in bovine kidney	
HS3G8	HS3G8 immunoreactivity is strong in bovine kidney and intestine	
AO4B08 and HS4E4	AO4B08 recognizes HS and heparin, interacting with ubiquitous, N-, 2-O-, and 6-O-sulfated saccharide units. HS4E4 preferentially recognized low-sulfated HS motifs containing idoA, N-sulfated and N-acetylated GlcN.	[238]

## 4. The Diverse Functional Properties of HS-PGs

### 4.1. Agrin

Agrin is a 400 kDa HSPG (212 kDa core protein), which interacts with low-density lipoprotein receptor-related protein-4 (LRP4) and α-dystroglycan. Chondrogenic signaling networks support chondrocyte differentiation and the upregulation of SOX9 and its transcriptional targets, COL2A1 and ACAN, which are major functional ECM components important for cartilage function. Agrin-induced chondrocyte differentiation does not induce chondrocyte hypertrophy [239]. LRP4 interacts with WNTs and BMPs [240,241,242] to regulate chondrocyte differentiation. Initiation of MuSK kinase in NMJ assembly requires neuronal agrin, which interacts with LRP, rapsyn and DOK-7 [243,244,245]. DOK7 promotes NMJ regeneration after injury [246]. Agrin clusters NMJ acetylcholine receptors [247]. NMJ agrin-Lrp4-MuSK cell signaling is disrupted in congenital myasthenic syndromes, Lambert-Eaton syndrome, Isaacs’ syndrome, Schwartz-Jampel syndrome, Fukuyama-type congenital muscular dystrophy, amyotrophic lateral sclerosis, and sarcopenia [248,249]

### 4.2. Perlecan

Perlecan (HSPG2) (400–467 kDa core protein) is a prominent multifunctional modular PG [250,251] of cartilages and vascular tissues, which provides cell-ECM communication [3]. HSPG2 promotes the formation of rudiment cartilages [252,253] and ossification centers in endochondral ossification and skeletogenesis [253]. Perlecan promotes tissue morphogenesis and has multifunctional roles in tissue stabilization [3] through interactions with HS binding structural glycoproteins and co-localizes with elastin in blood vessels [254]. Perlecan acts as a shear flow biosensor for endothelial cells, regulating vascular tone and blood pressure through feed-on effects on SMCs [255]. Perlecan also monitors shear flow in cannalicular fluid in bone, regulating osteocytic bone metabolism [256]. Perlecan provides mechanoresponsive and osmoregulatory properties to cells [2]. Interactive instructional cell-ECM links aid in the cellular orchestration of tissue homeostasis [3]. The multifunctional properties of perlecan aid wound healing and tissue repair processes [257].

### 4.3. Collagen XVIII

Collagen XVIII, long (187 kDa core protein), intermediate and short alternatively spliced isoforms form networks with perlecan in blood vessel basement membranes and also stabilize the ECM [258,259]. Collagen XVIII’s stabilizing roles in tissues become apparent when mutations lead to a collagen XVIII deficiency [259].

### 4.4. The Syndecan Family

The syndecan PG family (SDC1-4) performs various functions [27,29,32,260,261], including cell–cell and cell–ECM interactions, wound healing [262], growth factor receptor activation, matrix adhesion and MMP activation [263,264,265,266]. SDC2 and SDC4 may act synergistically in many of these processes. SDC-1 is a G-protein coupled co-receptor in cell proliferation/differentiation. SDC-3 is a receptor for HB-GAM, promoting neurite outgrowth and synaptic plasticity during neural network development [31,260,267,268].

### 4.5. The Glypican Family

The glypican PG family (GPC1-6) has multiple regulatory roles in cell signaling in tissue development and repair processes in health and disease [26,251,263,269,270]. GPCs regulate adherent cell signaling generated by shear flow and also have roles in tumor development. Investigations show GPCs may be of application in biomedicine [26,271,272,273].

### 4.6. Serglycin

SRGN, Serglycin, is a low-molecular-weight intracellular heparin-PG that can also be secreted and incorporated into the ECM. SRGN levels are elevated in inflammatory conditions [274,275]. IL-1β or TNF-α increase SRGN expression in vitro. SRGN stores key inflammatory mediators and proteases (elastase, chymase, tryptase, carboxypeptidase) as inactive forms in storage granules and secretory vesicles [276]. SRGN has multiple functional roles in a wide range of tissues in health and disease [277].

### 4.7. The Neurexins

Neurexins have important roles in synaptic stabilization and function. A vast array of synaptic protein interactions with the neurexin-α, β, γ family provide specificity in synaptic interaction [278,279,280] in pre- and post-synaptic interconnections that promote neurotransduction specificity and efficiency and neural synaptic plasticity [281]. Such interactions organize multi-protein pre-synaptic voltage-gated Ca channels and neuronal receptors [16,282,283] of importance in neural transduction.

### 4.8. Pikachurin

Pikachurin stabilizes the photoreceptor axenome primary cilium and ribbon sy napse and interacts with photoreceptor GPR179 and α-DG to facilitate the phototransductive process [284,285,286] and essential communication with bipolar retinal neural networks in visual processing. Vision is the primary human sense, sending ~60% of all inputs to the brain for processing [287,288,289].

### 4.9. Eyes-Shut

Interactions between Eyes-shut and matriglycan *O*-mannosyl glycan of α-DG stabilize the photoreceptor ribbon synapse and improve communication with the bipolar retinal neural network [290,291] required for high-quality visual acuity [292]. The importance of such interactions in ocular vision becomes apparent in mutations in *Eys*, which occur in retinitis pigmentosa, which displays vision impairment [293,294]

### 4.10. SPOCK

SPOCK-I-III are Kazal-like domain PGs with a modular structure, similar to perlecan and agrin, characterized by five domains [241,242,295,296]. Domain I is a SPOCK-specific N-terminal domain with no significant homology to other proteins other than members of the testican/SPOCK PG gene family. Domain II is a cysteine-rich module homologous to follistatin, also found in agrin. Domain III is homologous to the extracellular calcium-binding domain of SPARC and contains two Ca^2+^-binding EF-hand motifs [297]. Domain IV is a disulphide-stabilized thyroglobulin-like domain, harboring a CWCV tetrapeptide sequence [298]. The C-terminal Domain V is unique to the testican/SPOCK family and harbors two potential GAG attachment sites [298]. SPOCK3 contains two consecutive SGD triplets, attachment sites for HS also shared by perlecan and agrin. SPOCKs are almost exclusively expressed in the CNS and are primarily HSPGs.

## 5. HS Functional Motifs That Provide Tissue Functions

Sequencing HS samples has identified several protein interactive motifs (Figure 2). The FGFR and FGF-1, 2 binding sites in HS have been identified. 2-*O* and 6-*O* sulphation motifs and *N*-sulfation in HS are key functional determinants of these binding sites. An HS pentasaccharide, which provides the anti-coagulant activity of anti-thrombin (AT), has also been identified. 2-*O*, 6-*O*, 3-*O* and *N*-sulfation motifs provide this biological activity. 3-*O* and 2-*O* sulfation in IdoA is a key functional determinant in AT. Sulfation at the C3 position of glucosamine is a relatively rare, but biologically significant, feature of AT and a key determinant of its binding properties to procoagulant proteases [299,300,301,302]. Lipoprotein lipase is an adipocyte enzyme that cleaves fatty acids from circulating lipoproteins, degrading circulating triglycerides in the bloodstream [303]. Wnt cell signaling pathways have roles in embryonic development and in carcinogenesis. Wnt cell signaling pathways determine cell fate, proliferation, cellular migration and polarity, cell death and also regulate the homeostasis of adult tissues. Wnt proteins are secreted, lipid-modified glycoproteins with poor solubility. Their interaction with HSPGs improves their solubility and PGs, such as perlecan transport Wnt proteins, aiding in the formation of morphogen gradients that drive embryonic development [304,305]. 2-*O*, 3-*O*, 6-*O* and *N*-sulfate motifs in HS promote interaction with Wnt proteins [306]. Roundabout 1 (Robo1) is a cell surface axon guidance receptor. Its interaction with HS and members of the Slit protein family regulates the formation of neural networks [307,308,309]. Glypican-1 interacts with Robo-1 and Slit proteins during neural network formation. Glypican 3 and Glypican 5 also interact with the primary cilium on the cell surface, which acts as a mechanosensory cell signaling hub [310,311]. The VEGF-A dimer HS interactive module has been identified, and this promotes vascular processes that support tissue development and wound repair processes [312,313].

## 6. CS Chain Biodiversity

Approximately two in every seven CS chains in aggrecan are terminated in 4, 6 disulfated GalNAc. This varies with the age and cartilage type. Four in seven CS chains are terminated by 4-sulfated GalNAc, and one in seven CS chains are terminated in a GlcUA linked to 4-sulfated GalNAc (Figure 3). Non-reducing terminal 4,6-disulfated GalNAc residues are 60-fold more abundant than 4,6-disulfated GalNAc in interior regions of the CS chain [314].

CS chains terminated in 4-sulfated GalNAc predominate in aggrecan from fetal to 15-year-old knee cartilage, whereas in adult 22–72 year olds, 50% of the CS chains were terminated in 4,6-disulfated GalNAc. GlcUA-4 sulfated GalNAc disaccharides terminated 7% of CS chains in fetal to 15-year-old cartilage but fell to 3% in adults, whereas GlcUA-6 sulfated GalNAc represented 9% of the CS chains in fetal to 72 year olds. This disaccharide is recognized by MAb 3-B-3 (−) [315].

The distribution of 4- and 6-sulfated CS epitopes is variable along a CS chain in aggrecan and is influenced by the maturational status of the cartilage or the extent to which the cartilage was sampled from a high- or low-weight-bearing region [316]. Certain trends have been observed in the sulfation patterns of CS in aggrecan chains. C-4-S is more predominant in aggrecan from fetal and young articular cartilage and occupies a central region in the CS chain, whereas non-sulfated chondroitin is more predominant towards the linkage region. C-6-S has a predominant distribution towards the non-reducing terminus and is more abundant in mature cartilage to the detriment of C-4-S sulfation [316,317]. Graded partial digestions of CS chains with chondroitinase ABC or ACII reveal regions along the CS chain, where MAbs 6C3, 4C3 and 7D4 are the most immunoreactive [317]. MAb 6C3 reacts optimally with regions of CS chains towards the non-reducing terminus, and this reactivity is removed during early stages of chondroitinase digestion. Further digestion of the CS chain removes MAb 4C3 reactivity, and continued digestion removes reactivity to MAb 7-D-4, which provides information on the relative placement of these sulfation motifs along the CS chain. While the fine structures of specific epitopes identified by neoepitope MAb 4C3 and 7D4 have yet to be identified, the reactivity of these antibodies in a range of tissues undergoing morphogenetic transition displays subtly differential immunolocalization patterns and are of functional significance [318,319,320,321,322,323,324,325,326]. MAb 3-B-3 identifies a non-reducing terminal disaccharide in CS, consisting of GlcUA-GalNAc-6-sulfate. This is termed a 3-B-3 (−) epitope to distinguish it from the 3-B-3(+) stub epitope disaccharide attached to the linkage region, which is generated by exhaustive end-point digestion of CS chains by chondroitinase ABC [317]. As noted above, this non-reducing terminal 3-B-3(−) epitope occurs in approximately two in every seven CS chains; disulfated C-4,6-S and C-6-S GalNAc also occur as components in this non-reducing terminal disaccharide in CS chains [314,315].
Figure 3Diversity of structural organization of the CS side chains in aggrecan. The annotations used in (**a**–**e**) are explained in the accompanying table. Regions 2, 6, 8, 9 of CS chains detected by MAb’s 3-B-3(−), 4-C-3, and 7-D-4 were identified following graded partial digestion of the CS chains using chondroitinase ABC [317]. The structures depicted are examples of the many permutations in CS structure possible. The CS chains of aggrecan are dynamic structures and vary in structure in a spatiotemporal manner during tissue development and in tissue morphogenesis and during ECM remodeling during tissue repair responses and wound healing.
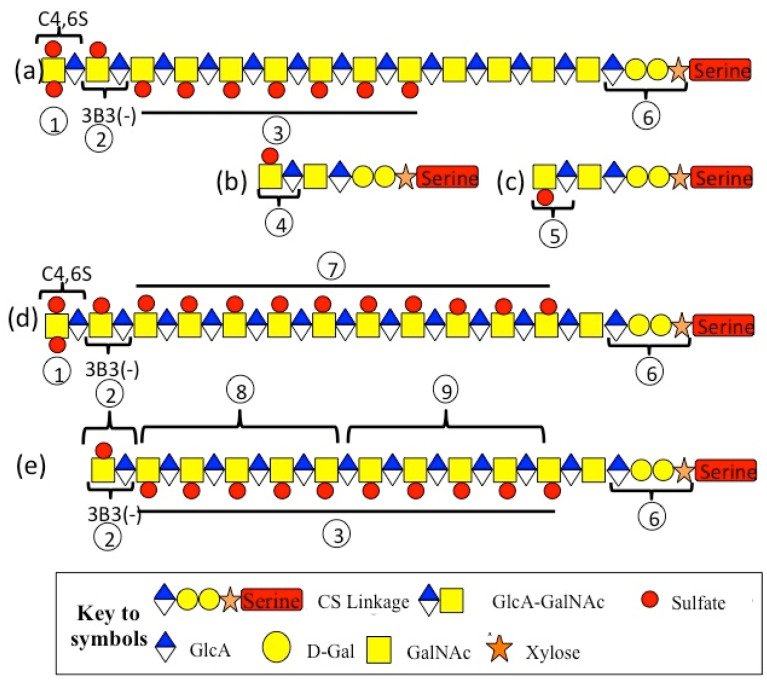

**Structural/Functional Diversity of the CS Side Chains of Aggrecan****Label****Structural/Functional Features of Annotated Regions****Refs.**1Non-reducing terminal disulfated CS epitopes interactive with morphogens such as IHH.[315,318,327]2New non-reducing terminus generated by HYAL4 digestion generates 3-B-3(−) epitope[328]3C-4-S epitopes predominate in foetal cartilage[319,329]4Reducing terminal 3-B-3 (+) stub epitopes attached to CS linkage region generated by chondroitinase ABC.[330]5Reducing terminal 2-B-6 (+) stub epitopes attached to CS linkage region generated by chondroitinase ABC.[330]6Linkage region to Serine residues on aggrecan core protein and GAG accepter region involved in the biosynthesis of CS chains.With ageing 6-sulfation levels in GalNAc increase in linkage region accompanied by increased Gal-6-S levels and lower 4-sulfation on GalNAc.[331]7Increased C-6-S epitope levels in overloaded cartilage regions and with ageing and reduced levels in OA are due to alterations in the expression of sulfotransferases[332,333,334,335]8Region of CS chain detected by MAb 4-C-3[317]9Region of CS chain detected by MAb 7-D-4[317]Note. Regions 2, 6, 8, 9 of CS chains detected by MAb’s 3-B-3(−), 4-C-3, and 7-D-4 were identified by graded partial digestion of CS chains using chondroitinase ABC [317].

## 7. Variation in the CS Chain Fine Structure in Development and Pathology

Aggrecan is a major structural PG of articular and other weight-bearing tissues and also has roles in the formation of embryonic neural networks and instructive roles over neural crest cells during the formation of the notochord, which is a precursor to other spinal tissues [127] (Figure 4a–c). Aggrecan from articular cartilage contains KS and CS side chains, which constitute ~90% of the mass of this PG. Aggrecan contains approximately 8 to 10 KS chains and ~100 CS chains; each GAG chain is spaced ~1 to 1.5 nm apart, and these range in size from 14 to 21 nm in length [117]. CS is the major aggrecan GAG and is organized into CS1 and CS2 regions. A key property of aggrecan is its interaction with HA via its G1 N-terminal globular domain. This process is important in the hydration and osmoregulation of tissues [336,337], and the hydrodynamic properties conveyed are important in weight-bearing tissues such as joint cartilages [338] and IVD [119,339], although it also hydrates and provides functional properties to soft tissues such as the brain [21,340,341].

Several years ago, MAbs 3-B-3(−) and 7-D-4 were shown to identify chondrocyte clusters in pathological (osteoarthritic) canine and human articular cartilage [342]. At this time, pre-dating knowledge of stem/progenitor cell niches in tissues, these cell clusters were considered a classical feature of the onset of late-stage degenerative joint disease and were interpreted to indicate a failed, late-stage response to replace PGs in a matrix extensively degraded by matrix proteases. An alternative explanation of this cellular phenomenon, however, has now emerged. It is now believed that these ‘chondrocyte clusters’ arise from adult stem/progenitor cell niches [343,344]. The 3-B-3(−), 4-C-3 and 7-D-4 CS sulfation motifs also occur in fetal development and are markers of anabolic processes in transitional tissues. An important feature of the stem/progenitor cell niche is the sulfation of the PG GAG side chains. The variable expression of GAG sulfotransferases and glycosyl transferases in stem/progenitor cell niches supports such an interpretation [345]. Cell clusters have also been shown to express Notch 1 and CD166, biomarkers that are synonymous with the stem cell niche [319,323,346].

**Figure 4 ijms-26-02554-f004:**
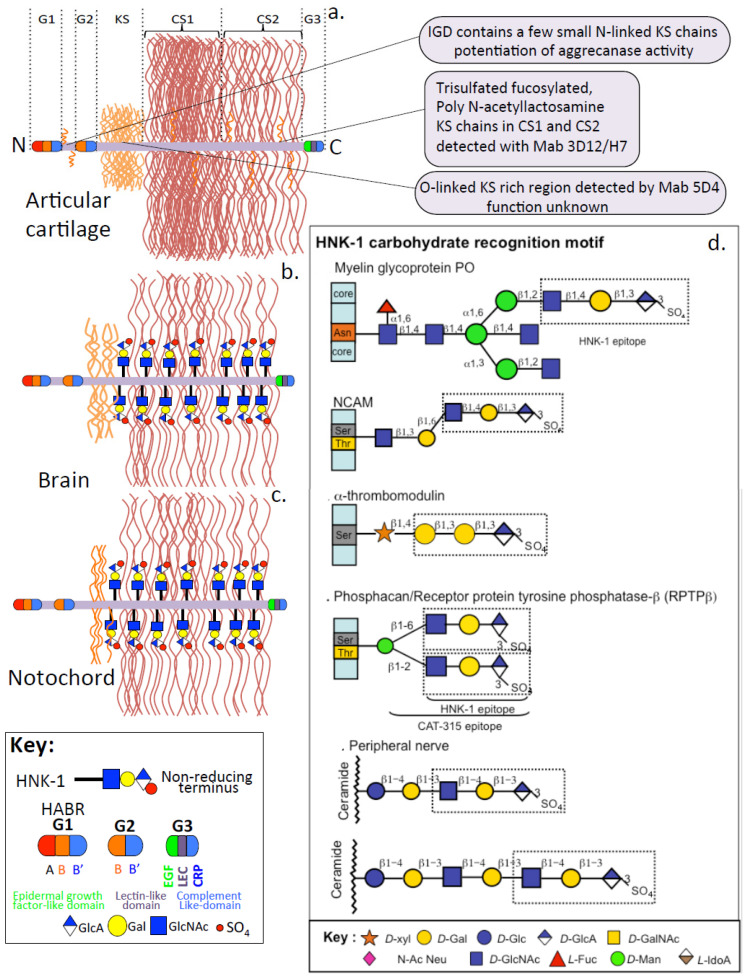
Schematic of aggrecans variable structure in articular cartilage (**a**), brain tissue (**b**) and notochord (**c**) and its N- and C-terminal globular domains interactive with HA and ECM components to form molecular networks. The G1 domain forms macromolecular complexes with HA stabilized by link-protein which hydrate tissues providing weight bearing properties to articular cartilages in joints and IVD. Aggrecan–HA complexes are also stabilized by brain link protein and tenascin R and C which are important in the hydration of brain tissue and the formation of perineuronal nets. Aggrecan G3 domains contain complement-like, EGF-like and lectin-like motifs which form networks in tissues forming an extended framework which conveys ECM mechanosensory cues to cells contributing to homeostasis. (**d**) The HNK-1 trisaccharide is a notable modification in aggrecan in brain and notochordal aggrecan with roles in the regulation of neural crest progenitor cells and nerve development but also occurs in embryonic cartilage aggrecan but absent in adult cartilage. HNK-1 has roles in the myelination of axons, and occurs in peripheral nerves and modifies CS substitution in α-thrombomodulin and phosphacan. Figure modified from [127] with permission.

### 7.1. Aggrecan CS Side Chain Modifications in Specific Tissue Contexts

Aggrecan is widely distributed in the articular hyaline cartilages of diarthrodial joints but also occurs in the elastic and fibrocartilages of rib, nasal and tracheal cartilages, larynx, outer ear and the epiglottis [56,347,348,349], and is an important functional component of the myocardial ECM [350]. Aggrecan has important roles in fetal heart development and is a functional ECM component, which contributes to the resilience of the endocardium, myocardium, epicardium and valve leaflets of mature heart tissue [351,352]. Aggrecan is also found in the CNS and PNS in perineuronal net (PNNs) structures (Figure 5). These are aggrecan–HA–tenascin C aggregate structures, which localize around neurons during development and are specialized forms of neural ECM, which have neuroprotective roles, control synaptic plasticity and have roles in memory [126,353,354].

### 7.2. Role of IHH in Chondrogenesis

Indian hedgehog (Ihh) [355], a member of the hedgehog protein family along with sonic hedgehog (Shh) [356], regulates chondrocyte differentiation, proliferation and maturation in articular cartilage development [357] and during endochondral ossification through interactions with parathyroid hormone-related peptide (PTHrP) [358] and BMP-mediated cell signaling [359]. Ihh has multiple functions during skeletogenesis [360,361,362]. Mice lacking the Ihh gene exhibit severe skeletal abnormalities, including markedly reduced chondrocyte proliferation and abnormal maturation and an absence of mature osteoblasts, which has detrimental effects on bone development [363]. Ihh and its receptor, smoothened (smo), are expressed in chondrocytes and osteoblasts; thus, Ihh may have a direct effect on osteoblasts, or its effects may be mediated indirectly through chondrocytes during the process of endochondral ossification. IHH colocalizes with aggrecan in the growth plate. Aggrecan regulates the expression of growth factors and signaling molecules during cartilage development and is essential for proper chondrocyte organization, morphology and survival during the formation of the axial skeleton. The sulfated GAGs of the CS and KS side chains of aggrecan provide water-imbibing properties, creating a large hydrophilic molecule important for the hydration of cartilage and the provision of its hydrodynamic weight-bearing properties, but also bind growth factors and morphogens crucial to chondrocyte maturation and function [327,364]. Thus, aggrecan should not be considered merely as a space-filling bulking ECM component that provides hydration and weight-bearing properties to tissues but also has cell instructive properties capable of modulating the activity of growth factors and morphogenetic proteins, thus mediating tissue development. Indeed, aggrecan knock-out mutants display a range of severe ECM defects, which support this proposal [365,366].

### 7.3. HNK-1 Trisaccharide

The human natural killer-1 (HNK-1) trisaccharide (HSO_3_-3GlcAß1-3Galß1-4GlcNAc-) is highly expressed in the nervous system on *N*-linked and *O*-mannose-linked glycans, and its spatiotemporal expression is strictly regulated [367] (Figure 4d). Monoclonal antibody 6B4 detects *O*-mannose-linked HNK-1 carried by phosphacan [368]. Mice deficient in the enzyme glucuronyltransferase (GlcAT-P), a key HNK-1 biosynthetic enzyme in GlcAT-P (B3gat1), display almost complete disappearance of the HNK-1 trisaccharide epitope in the brain. This is accompanied by a significant reduction in long-term potentiation and dysfunctional spatial learning and memory formation, demonstrating HNK-1’s physiological roles in higher brain function.

HNK-1 trisaccharide is also an autoantigen associated with peripheral demyelinative neuropathy and has important roles in myelination processes and the preservation of neuronal signaling efficiency [369]. Structurally distinct HNK-1 epitopes are expressed in specific proteins in the nervous system. The HNK-1 epitope on AMPA receptor sub-unit GluA2 and aggrecan regulate neural plasticity but in different ways [370]. The HNK-1 epitope is indispensable for the acquisition of normal neuronal function, cognitive learning and memory [371]. Despite the widespread disappearance of HNK-1 in the GlcAT-P KO mouse brain, it remained in specific regions, such as perineuronal nets (PNNs) [372]. HNK-1 in PNNs appeared to be synthesized by a unique biosynthetic pathway. Loss of HNK-1 alters the distribution of postsynaptic proteins, such as alpha-amino-3-hydroxy-5-methylisoxazolepropionate (AMPA)-type glutamate receptor GluR2 and PSD-95 (postsynaptic density protein 95), also known as SAP-90 (synapse-associated protein 90), from spine heads. GluR2 is a major HNK-1 carrier glycoprotein that promotes spine morphogenesis [370]. The overexpression of GluR2 promoted spine growth in both wild-type and GlcAT-P-deficient neurons, but the increase in GlcAT-P-deficient neurons was lower than that in wild-type neurons. HNK-1 is, thus, a key factor for normal dendritic spine maturation and is involved in the distribution of postsynaptic proteins.

A novel truncated form of phosphacan, phosphacan short isoform (PSI), representing the N-terminal carbonic anhydrase- and fibronectin type III repeat domains and half of the spacer region, is modified with HNK-1 trisaccharide and oligosaccharides but no GAG. PSI interacts with the Ig cell adhesion molecules F3/contactin and L1 and promotes the outgrowth of cortical neurons [146]. Phosphacan interacts with Ng-CAM and NCAM and modulates neuronal and glial cell adhesion, neurite outgrowth, and signal transduction during CNS development [373]. Instructional interactions between neurons and glial cells regulate the development and regeneration of the CNS [145]. Migrating neurons are guided by radial glial pericellular scaffold interactions that direct the growth and migration of axons during the development of neural networks. In adult neural tissues, astrocyte PGs and myelin glycoproteins inhibit neuroregeneration. The CSPG splice variant of RPTP-β (PTP-ζ), DSD-1/phosphacan, glial PG is developmentally regulated and can display stimulatory or inhibitory properties over neurons via interactions with IgG family neuronal receptors [143]. DSD-1 glial cell CSPG, the mouse homolog of phosphacan, promotes neurite outgrowth in rat embryonic mesencephalic (E14) and hippocampal (E18) neurons. This contrasts with other CSPGs, such as members of the lectican family, which inhibit neuronal outgrowth and regeneration. However, DSD-1 also displays inhibitory properties in DRG neuron cultures, and this property appears to be associated with the DSD-1 core protein. Thus, the stimulatory or inhibitory properties of DSD-1 are neuron lineage dependent in specific tissue contexts [144]. The 3-O-sulfation of the terminal GlcA of HNK-1 trisaccharide acts as an inhibitory signal for the initiation of CS biosynthesis on thrombomodulin, preventing the interactive properties of α-thrombomodulin with thrombin and reducing its anti-coagulant activity [374,375]. HNK-1 trisaccharide is expressed on aggrecan in early cartilage development; however, by embryonic day 14, it is no longer detectable [376]. HNK-1 trisaccharide is also expressed on notochordal aggrecan and may have instructive roles over neural crest cell migration during the maturation of the notochord and formation of neural networks prior to the development of cartilaginous embryonic spinal skeletal tissues.

**Figure 5 ijms-26-02554-f005:**
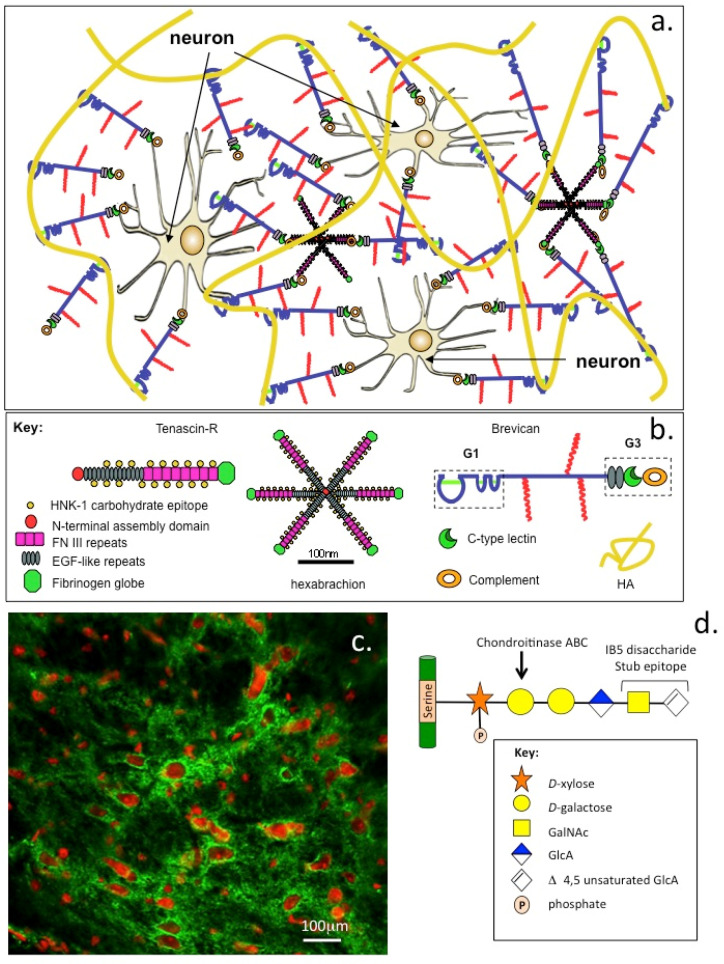
Schematic depiction of HA-lectican molecular networks in perineuronal nets showing ternary complex formations stabilized by tenascin-R. Only brevican is depicted for clarity but such networks also contain aggrecan, a major component and neurocan to a lesser degree. Phosphacan is also a component of perineuronal nets (**a**). Key showing the structural organization of some perineuronal net components (**b**). Fluorescent immunolocalisation of perineural nets in the cerebellum. Neuronal nuclei are visualized using propidium iodide (red) and aggrecan using a combination of chondroitinase ABC and FITC labeled Mab 1B5 (green) to a sub epitope which remains attached to the CS attachment region to aggrecan core protein (**c**). Schematic structure of the 1B5 stub epitope terminated in the unsaturated Δ-4,5 uronate residue which absorbs strongly at 232 nm (**d**). Figure modified from [377] with permission.

## 8. The Structural and Functional Diversity of KS, HS and CS-PGs

The biodiversity of PG GAG side chains conveys a wide range of interactive properties with a large range of ligands such as growth factors, cytokines, morphogens, cell surface receptors and structural ECM glycoproteins, Cell associated and ECM PGs which are decorated with these GAGs thus display a wide range of functional atttributes to cell. These are summarized in Table 3.

## 9. Interactive Properties of GAGs and How They Influence Tissue Development

Some of the earliest studies on glycans were on mammalian corneas and lenses, and these uncovered key cell interactive concepts fundamental to our understanding of basic cell biology. PGs and their GAG side chains are essential components of the ECM of the lens capsule and basement membranes. Lumican, fibromodulin and keratocan KSPGs organize collagen fiber diameters and strategic spacing in the corneal stroma to maintain visual clarity. Genetically engineered mice and gene mutations that spontaneously arise show key aspects of PG biosynthesis and the roles these play in signal transduction to regulate cell signaling and the avascular and immune status of tissues [380]. Furthermore, control of ECM signals by neurons is pivotal to brain development, plasticity, and repair and axonal guidance via receptor–ligand interactive crosstalk with ECM components [20,21]. The interaction of semaphorin-5A (Sema5A) with HS and CS provides bifunctional attractive and inhibitory properties, affecting neuronal growth, and represents a molecular switch in neural development [425]. A unique feature of class 5 semaphorins is their seven extracellular thrombospondin-1 (TSP-1) repeats, which have GAG interactive properties that convey axonal guidance and shape the nervous system during development, neuronal proliferation and migration in neuritogenesis and synapse formation [426]. HS-GAG binding is preferred over CS-GAG and mediates Sema5A oligomerization. Such interactions regulate Plexin-A2-dependent neuroprogenitor cell migration in developmental and neurological disorders and show how PGs regulate such processes. Robo–Slit interactions regulated by HSPGs also provide neuronal guidance cues during innervation of the lens and cornea [427]. Plexin A1 has also been demonstrated to be a receptor for Robo interactions that regulate neuritogenesis [428,429]. Robo–Slit interactions with GAGs provide axonal guidance in neural network assembly in a number of tissue contexts in the brain and are also operative in the spinal cord [430,431,432,433]. Furthermore, neural tissue homeostasis and repair are regulated by CS and DS PG motifs [434], and a number of KSPGs also modulate the activity of a range of neuroregulatory proteins in the brain [59], which is consistent with the unique functional capabilities of KS [47] and its roles as an electrosensory neurosentient bioresponsive cell instructive GAG [109]. Lumican and keratocan have been immunolocalized in the rat spinal cord and IVD [82].

### 9.1. Neural GAG Structures Have Significant Functional Roles in Synaptic Activity

#### HSPG-Specific Roles in Synaptic Stabilization, Specificity of Interaction and Plasticity

As already discussed, the neurexin HSPG family has key roles to play in synaptic stabilization through interactions with a vast array of adhesive synaptic proteins [15], which also ensure specificity and precision in such synaptic interactions and synaptic plasticity. HS interactions extend the range of interactive ligands operative in such interactions [16,19]. Interaction with the highly glycosylated glycoprotein DG also contributes to the stabilization and functional status of synaptic structures in cell signaling processes that regulate cellular behavior [13,435,436]. Figure 6 schematically illustrates some of these proteins, such as the neuroligins [403,437] and LRRTMs [438,439], and adaptor proteins, such as MINT [440] and CASK (calcium/calmodulin-dependent serine protein kinase) [441]. CASK interacts with other proteins through its multi-modular domains and is involved in memory formation, neurotransmitter release, cell adhesion and pre- and postsynaptic signaling. Munc18-1 is a neuronal protein that interacts with syntaxin 1 and is required for synaptic vesicle exocytosis mediated by the Mint1 and Mint2 adaptor proteins [442]. Mints 1–3, also referred to as X11α/β/γ, X11/L1/L2, or APBA1/2/3, multidomain adaptor proteins, are interactive with a variety of synaptic proteins, such as CASK. CASK is a scaffold protein, with roles in the development of the nervous system and the release of neurotransmitters [441]. In the brain, Mint proteins constitute part of a multimeric complex containing Munc18-1 and syntaxin, which function as intermediates in synaptic vesicle docking/fusion. This process appears to be mediated by a phosphotyrosine-binding domain in Mint that specifically binds to phosphatidylinositol phosphates groups [443]. Neurexin-neuroligin synaptic interactions facilitate the recruitment of AMPA and other glutamate receptors such as NMDA and GluD2, which are important for neurotransmitter transmission and neuronal function [403,437]. Neurexin-3 has important roles in synapse development and synapse functions through interactions with leucine-rich-repeat transmembrane neuronal proteins (LRRTMs) [444]. Figure 6 shows a schematic presentation of these proteins.

### 9.2. DG-HSPG Interactive Roles in NMJ Assembly and Neuromuscular Regulation in Health and Disease

DG also has roles in the assembly of the neuromuscular junction (NMJ) and synaptic basement membrane [445], where, along with perlecan, it clusters acetylcholinesterase in the NMJ, an essential feature that facilitates neuromuscular control by motoneurons [445,446]. DG has widespread properties in the development and function of the nervous system [435,436]. DG-HSPG interactions provide synaptic plasticity and specificity [13]. DG is also a component of dystrophin glycoprotein complexes, with roles to play in skeletal tissue dynamics [14] and tissue stabilization (Figure 7). Furthermore, DG also serves as a cell signaling platform and is, thus, a multifunctional glycoprotein, interactive with a wide range of proteins [447]. DG also maintains the integrity of the inner retinal membrane [448] and has roles in the formation of the photoreceptor ribbon synapse [289], connections between photoreceptors and retinal bipolar neurons, ensuring effective communication between them [288] and binding of the orphan receptor GPR179 to DG-pikachurin complexes essential for photoreceptor organization and function [285]. DG is of particular importance in the dynamic stabilization and function of the ECM [449].

DG is composed of an extracellular α-domain, which is heavily glycosylated, and this is highly interactive with a wide range of ECM components, including HSPGs through interactions with their Lam G core protein motifs [450] (Figure 7). DG also contains an intracellular β-domain, which connects to the cytoskeleton, and, thus, collectively, the α- and β-domains provide a transmembrane cytoskeleton-ECM direct link operative in cell–ECM communication and the mechanical regulation of connective tissue cells. Thus, DG provides both ECM stabilization and is a cell signaling platform through which cells can be regulated (Figure 8).

DG is a novel laminin and agrin receptor. A drastic reduction in DG in muscle tissue, caused by an absence of dystrophin, leads to muscle cell death and the symptomatology of Duchenne muscular dystrophy, which is caused by a mutation in the X-linked dystrophin gene [451]. However, DG also has roles in cellular differentiation processes in many different cell types and is now of interest in the development of many diseases [452,453]. Inappropriate glycosylation of DG appears to be a central event in the pathogenesis of several complex diseases [454,455]. Improvements in glycomics methodologies suggest that glycosylation could potentially be modulated in DG samples to ameliorate the pathological progress of these diseases. It is, thus, essential that the mechanism of action of DG in disease processes be fully deciphered in order to develop potential therapeutics. Glycans and GAGs clearly have key roles to play in these biological processes.

**Figure 6 ijms-26-02554-f006:**
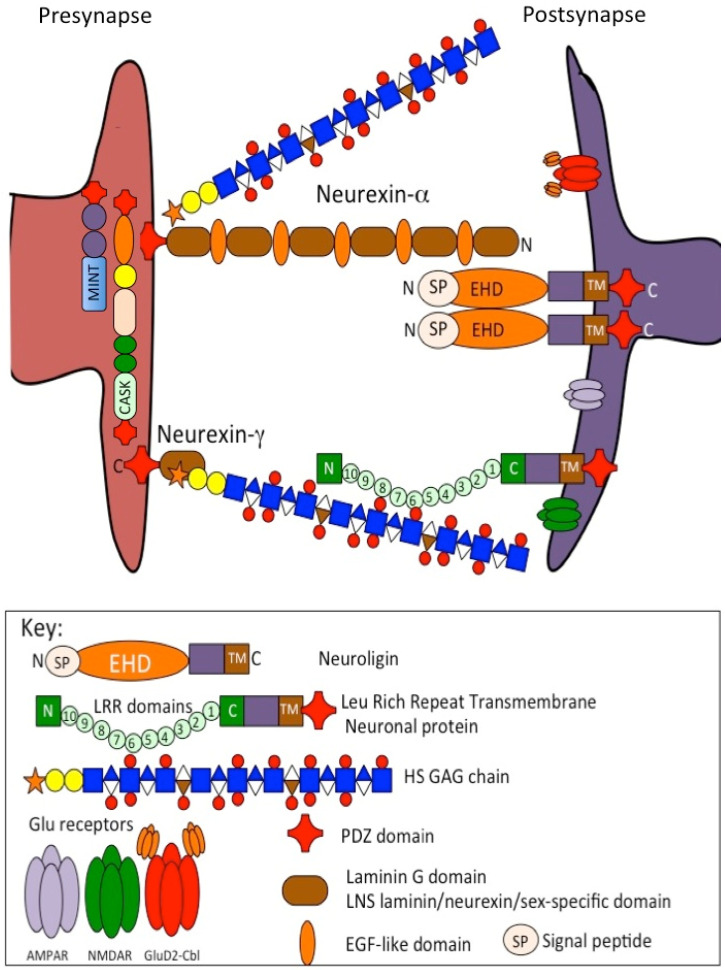
Schematic depiction of a neuron synapse depicting the stabilizing role played by the HSPG neurexin. which interacts with a large number of synaptic proteins providing synaptic plasticity and specificity of synaptic interactions. Neurexin-α and neurexin-γ isoforms are depicted interacting with neuroligin and leucine rich repeat transmembrane glycoproteins. The glutamate receptors AMPAR (α-amino-3-hydroxy-5-methyl-4-isoxazolepropionic acid receptor) and NMDAR (*N*-methyl-D-aspartate receptor) and GluD2 (glutamate dehydrogenase-2)-Cbl (cerebellin) are also potential ligands for the neurexin family. Abbreviations used: EHD, esterase homology domain; PDZ, a term derived from post-synaptic density protein (PSD95); drosphila disc large tumor suppressor (Dig 1) and zona occludens-1 protein (Zo-1); LNS, laminin, neurexin, sex-hormone binding globulin; CASK, calcium calmodulin-dependent serine protein kinase-3; TM, transmembrane; LRR, leucine-rich repeat; SP, signal peptide; MINT, molecular interaction; LRRTMs, leucine-rich repeat transmembrane neuronal proteins.

**Figure 7 ijms-26-02554-f007:**
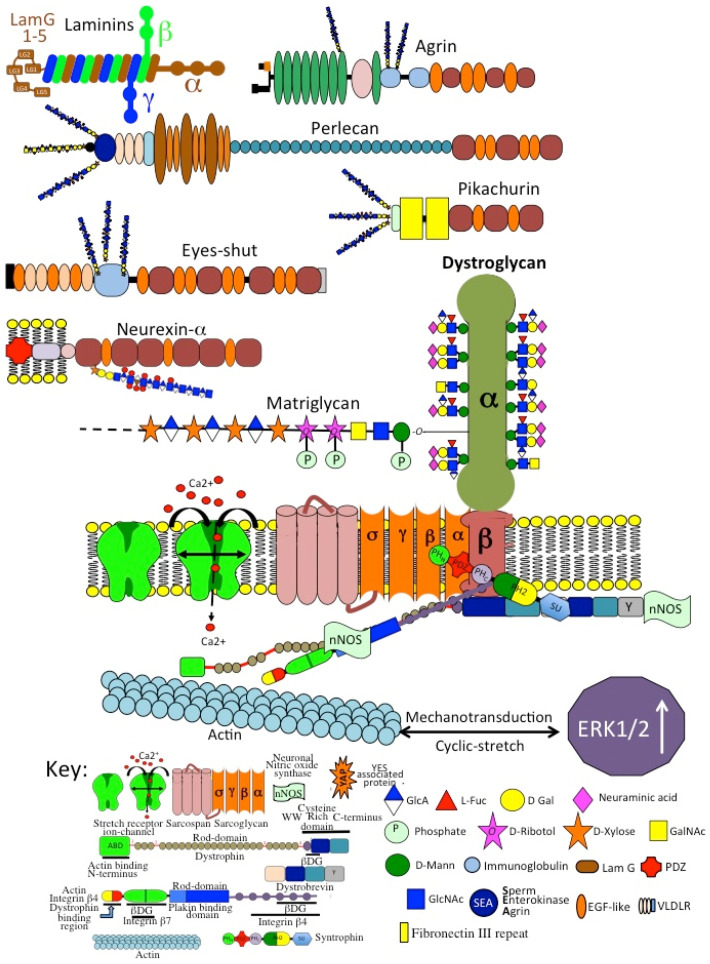
Schematic depiction of key interactive glycan structures in dystroglycan, extracellular and intracellular cell signaling structures and some of the HSPG effector PGs that interact with dystroglycan. Laminin also interacts with DG through its LamG 1–5 domains. Laminin G domains in the core proteins of agrin, perlecan, Eyes-shut, pikachurin and the neurexins interact with DG providing ECM stabilization and cell-ECM communication in cell signaling pathways that regulate tissue homeostasis [436]. DG acts as a transmembrane linkage glycoprotein between the ECM and cytoskeleton and has particularly important roles to play in the nervous system where interactive HSPGs act as effector molecules in many neural processes [13,435].

**Figure 8 ijms-26-02554-f008:**
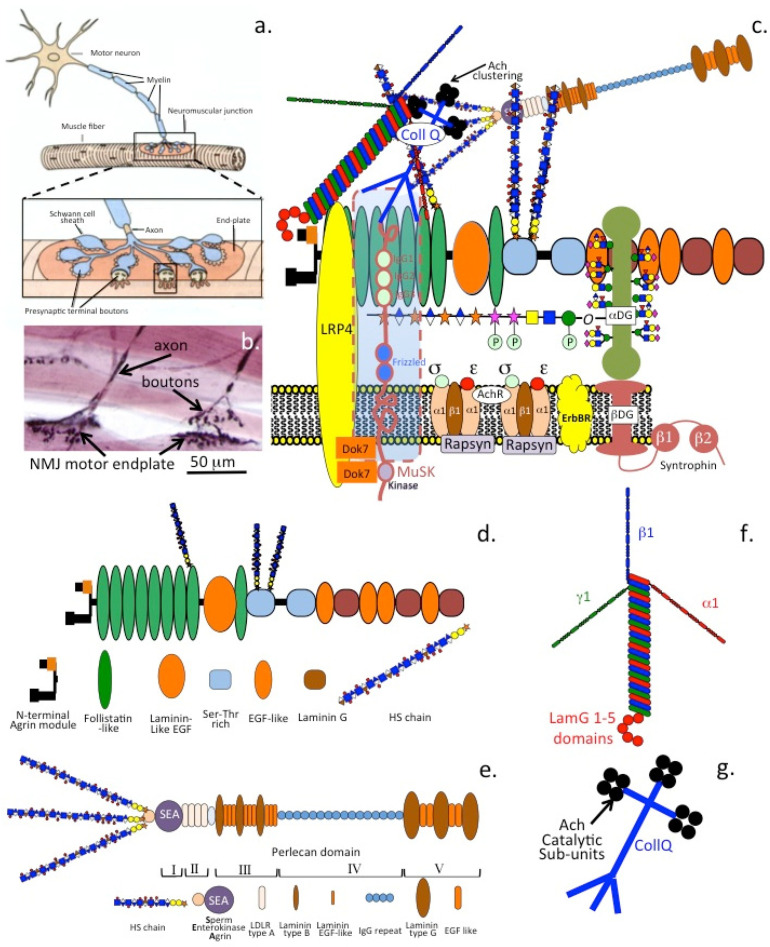
The complexities of the NMJ basal membrane showing a diagram of the NMJ (**a**), immunolocalisation of nerves in a NMJ using H&E staining of muscle fibres and silver esterase staining of nerves (**b**). Simplified schematic of key functional components of the NMJ including laminin, perlecan, agrin, dystroglycan, LRP4 and MuSK glycoproteins (**c**). Cholinesterase receptors are localised and anchored in the NMJ basement membrane by agrin, Musk and dystroglycan. Perlecan interactions with DGand collagen Q localise active cholinesterase sub-unit clustering, Cholinesterase and its receptors are key components that provide motor activity in the NMJ. Rapsyn adapter protein [456] and ErbR transmembrane tyrosine kinase receptors [457] also have roles in the stabilisation of cholinesterase receptors. Cytoskeletal connections provided by the beta sub-unit of DGcontribute to cell-ECM communication and neuronal cell signalimg pathways. Details of the structural organisation and domain organisation of agrin (**d**), perlecan (**e**), laminin (**f**), and collagen- Q cholinesterase clusters (**g**) are also provided in schematic diagrams.

## 10. Conclusions

The aim of this study was to convey the complexities of GAG’s fine structure and how this impacts tissue function in specific cell and tissue contexts in health and disease. The roles of specific GAGs in neural issues were also covered, and some GAG isoforms which have not previously been considered to have roles in this context were outlined. KS is an underappreciated GAG in tissue function in this respect, and low-sulfation KS has emerging roles in neural tissues. Further studies in this area are warranted. HS, in all its forms, has a multitude of interactive properties with a myriad of effector proteins, which convey important functional properties to tissues. This becomes apparent when abnormal assembly or degradative processes result in GAG dysfunction in a range of human diseases. A greater understanding of how GAGs convey properties to tissues and instruct cellular behavior and advances in GAG analytical techniques offers exciting possibilities, coupled with advancements in glycomics research in the development of GAG biotherapeutics. GAGs certainly have a diverse range of cell instructive properties, which could be potentially harnessed to treat functional deficits in tissues. Thorough basic studies on the roles of GAGs in tissues may well be insightful and are essential to take the field of GAG biotherapeutics forward.

## Figures and Tables

**Figure 2 ijms-26-02554-f002:**
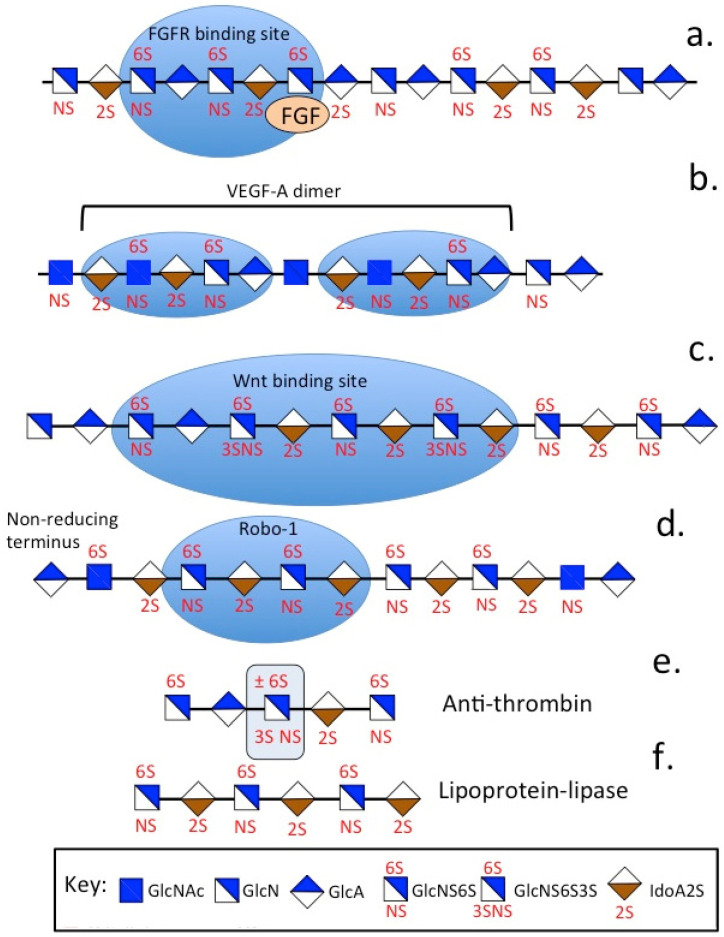
Specific interactive glycan sequences in HS that have been identified. Identification of specific GAG sequences interactive with fibroblast growth factor and fibroblast growth factor receptor (**a**), vascular endothelial growth factor A dimer (**b**), wingless-type MMTV integration site family cell signalling complex (Wnt) morphogens (**c**), Roundabout-1 axonal guidance receptor (Robo-1) (**d**), antithrombin (**e**) and lipoprotein lipase (**f**). Glycan residues are depicted using standard SFNG icons as indicated in the key.

**Table 3 ijms-26-02554-t003:** The structural biodiversity of KS, HS and CS PGs.

PG	Structural and Functional Features	Core Protein Size (kDa)	GAG Chains Present
KS-PGs			
Aggrecan, CSPG1 (ACAN)	Hydrates and provides hydrodynamic viscoelastic weight bearing properties to cartilages [57].	208–220	CS, KSI/KSII
Lumican (LUM)	MMP inhibitor, anti-angiogenic anti-cancer agent, regulates regularly organized slender collagen fibrils in cornea [378].	38	KSI
Keratocan (KERA)	Essential ECM component of the lens capsule, organizes collagen fiber diameters and spacing in the corneal stroma to maintain stromal clarity [194,198,379,380].	37–50	KSI
Fibromodulin (FMOD)	Cell regulatory multifunctional matricellular modulator, maintains cellular architecture for normal tissue function, regulates collagen fibrillogenesis [381,382,383].	42	KSI
PRELP (Prolargin)	PRELP is an anchoring component in many basement membranes binds type I and II collagens and perlecan to stabilize the basement membrane [188,198].	44	KSI
Osteomodulin (osteoadherin)	OMD, is a KS-SLRP that binds to osteoblasts via αςβ3 integrin and regulates osteogenesis through its interaction with BMP2. WNT1 transcriptionally activates expression of OMD [195,196,384]	42	KSI
Osteoglycin(OGN) (Mimecan)	OGN, is a class II SLRP with diverse roles in ECM assembly, regulates bone formation along with TGF-β1/TGF-β2 that controls collagen fibrillogenesis and has glucose regulatory roles in metabolic health, cancer and diabetes [199,200,385,386,387,388,389]	35	KSI
Chondroadherin (CHAD)	CHAD, is a 38 kDa member of the KS-SLRP family containing 11 LRRs that bind to α2β1 integrin, type I, II and VI collagen and has an anchoring role in ECM stabilization, binds cells to the ECM and mediates cell-ECM communication through interactions with cell surface PGs such as the syndecans [390,391,392,393,394]	36–38	KSI
Claustrin	Claustrin is an anti-adhesive neural KS-PG [395]	105	KSII
Synaptic vesicle PG (SV2)	SV2 is a synaptic vesicle neurotransmitter transporter and smart storage PG, SV2A, SV2B, SV2C paralogs share 60% sequence and 80% structural homology. SV2A controls transmitter release, SV2B is the primary paralog expressed in the retina, SV2C has roles in synaptic plasticity [29,30,31,32,33,34,35].	Occurs as H 250 kDa and L 100 kDa forms	KS
Podocalyxcin (PODXL,TRA-1-60)	Transmembrane, anti-adhesive sialo-KS-PG, up-regulated in many cancers and is a tumor stem cell biomarker [164,396]	65	KS
Phosphacan	Soluble ectodomain of RPTP-ζ exists as three splice variants, roles in perineuronal net assembly and function in cognitive processes, modulates neurite extension in formation of neural networks [143,144,145,146,148].	300	KS, CS, HNK-1
HS-PGs			
Agrin	400 kDa HSPG, interacts with LRP4 and α-DG. Promotes chondrocyte differentiation, upregulates SOX9, COL2A1, ACAN [239]. Activates MuSK in NMJ, interacts with rapsyn, LRP,DOK, clusters Ach receptors in NMJ neuromuscular control [397]	212	HS
Perlecan (HSPG2)	Multifunctional, modular HS/CS PG, interacts with growth factors, controls cell proliferation and differentiation, cell signaling and tissue morphogenesis, facilitates cell-ECM communication, shear flow biosensor important in tissue homeostasis and function [3,250,257,398,399].	400–467	HS/CS
Collagen XVIII	Stabilising, basement membrane component in laminin, nidogen HSPG networks [251,263,400].	187	HS
The syndecans	SDC 1–4 are G-protein coupled co-receptors in cell proliferation and differentiation, regulating growth factor interactions, tissue development, wound repair, tissue regeneration, inflammation in health and disease [23,24,31,32,260,261].	22–48	HS/CS
The glypicans	GPC1-6 have multiple regulatory roles in cell signaling in tissue development and repair processes in health and disease [26,251,269,270].	62	HS
Serglycin	Intracellular heparin PG storing bioactive compounds in vesicles [276] in immune [401] and neuroendocrine cells. With varied roles in health and disease [277].	17.3	Heparin
Neurexin (NRXN)	NRXN1-3 [402] act as receptors and cell adhesion molecules [19] aiding in synaptic development [403] and stabilization and signaling along with a vast collection of ligands [15,16]. LamG motifs interact with α-DG stabilizing synaptic activity. NRXN3 provides synaptic plasticity.	n/a	HS
Pikachurin	Pikachurin has roles in synaptic assembly [404] interacting with α-DG in photoreceptor ribbon synapse assembly [285,289] facilitating interaction with retinal bipolar neural networks in visual processing [288].	n/a	HS
Eyes-shut	Eyes-shut stabilises the photoreceptor primary cilium axenome which connects the inner and outer regions of the photoreceptor and has essential roles to play in phototransduction [291], Eyes shut deficiency leads to autophagy of photoreceptors and impaired vision [405].	n/a	HS
SPOCK (testican, sparc/osteonectin, cwcv and kazal-like domains PG, SPARC (osteonectin)	SPOCK-2 is induced by viral infection or IFN, and is secreted to the ECM, where it blocks virus-cell attachment and entry. SPOCK regulates malignant tumor development [406] and has roles in embryonic development [407] and neuromuscular tissue development [408].	48.4	CS/HS
CS-PGs			
Aggrecan	Hydrates and provides hydrodynamic viscoelastic weight bearing properties to cartilages [57,118] but is also a component of heart and brain tissue [125]. HNK-1 in heart and brain aggrecan provides additional interactive properties [372].	208–220	CS, KS
Versican (PG-M, CSPG2)	Versican plays diverse roles in cell adhesion, proliferation, migration and angiogenesis and is so named in recognition of its versatile modular structure [409]. Versican has key roles in inflammation through interactions with adhesion molecules on the surfaces of inflammatory leukocytes and chemokines that recruit inflammatory cells [410]. Versican forms macromolecular complexes with HA which are looser than aggrecan-HA aggregates conducive to cell attachment and migration [411].	265	CS
Neurocan	Neurocan modulates cell adhesion and migration in brain development and has roles in the formation of perineuronal nets and their functional interactive properties [412,413,414,415,416].	145	CS
Brevican (BEHAB,CSPG7)	Brevican is localised to the surface of neurons in the brain and maintains molecular networks around neurons which may slow brain ageing and AD development [416].	96	CS
Decorin (DCN)	Widely distributed and highly interactive forming multifunctional networks [417]. DCN has roles in tissue protection [418] and wound repair, angiogenesis, tumor metastasis [419], autophagy, immune regulation and inflammatory diseases [420]. DCN has antifibrotic, anti-inflammatory, antioxidant, antiangiogenic and onco-suppressive properties [421] and inhibits TGFβ activity [422].	36	CS/DS
Biglycan (BGN)	BGN is both a structural ECM component and a signaling molecule [423]. BGN LRRs have interactive properties with a range of protein ligands contributing to ECM stabilization and function. When proteolytically released from the ECM, biglycan acts as a danger signal of tissue stress or injury. Biglycan links innate immunity receptors and activators of the inflammasome, stimulating multifunctional proinflammatory signaling in tissue damage [423].	38	CS/DS
Asporin (ASPN)	ASPN contains a distinctive group of N-terminal D-Asp-residues which are linked to cancer progression and OA. Regulates TGFβ, Wnt/β-catenin, notch, hedgehog, EGFR, HER2 cell signaling pathways [424].	42	CS

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
