# Peer review of "Glycosaminoglycans, Instructive Biomolecules That Regulate Cellular Activity and Synaptic Neuronal Control of Specific Tissue Functional Properties"

_ijms, 2025, doi:10.3390/ijms26062554_

Round 1

Reviewer 1 Report

Comments and Suggestions for Authors

The manuscript by Prof. Melrose is a good attempt at covering a wide-ranging topic. However, in my opinion, the manuscript can benefit from a narrower scope. For starters, the title itself can be perplexing to readers. For instance, what is “Homeostatic Cell Instructive Capability”? The convoluted title gave the impression that the author is trying to cover too-expansive a topic for a single review. Although I saw nothing wrong in the scientific content and the literature citations are comprehensive, the manuscript is diffusive and lacks the focus needed to grab the reader. I recommend that the author pare down the scope of the review. For instance, I would remove the entire section 7 (HS-binding proteins in health and disease), which I don’t think serves an essential purpose. If possible, I would also remove the sections on basement membranes and structural tissues and narrow the focus to the functions of GAGs in sensory tissues. In addition, what’s the author’s definition of sensory tissue? There are extensive discussions of the role of proteoglycans in the synapse. Can cells in the CNS really be considered sensory tissue? I think it benefits the manuscript to focus on proteoglycans in tissues that sense the outside environment directly. In summary, I think the manuscript in its current form is just another general GAG review in disguise. The readers would be much better served with a review that focuses on how GAGs help the tissues to sense the environment.

Author Response

Manuscript ID: ijms-3442013 entitled ‘’ Glycosaminoglycan Components of Cell Associated and ECM Proteoglycans Convey Molecular Recognition, and Homeostatic Cell Instructive Capability to Structural and Sensory Tissues

Point by point responses to reviewer 1.

The manuscript has been extensively re-written in a revised manuscript. All new segments are highlighted in yellow.

Reviewer 1

Reviewer comment

For starters, the title itself can be perplexing to readers. For instance, what is “Homeostatic Cell Instructive Capability”? The convoluted title gave the impression that the author is trying to cover too-expansive a topic for a single review.

Author response

A new more focused title without the offending terms has been provided which hopefully will be more acceptable.

New Title: “Glycosaminoglycans, Instructive Biomolecules That Regulate Cellular Activity And Synaptic Neuronal Control Of Specific Tissue Functional Properties”.

Reviewer comment

I recommend that the author pare down the scope of the review. For instance, I would remove the entire section 7 (HS-binding proteins in health and disease), which I don’t think serves an essential purpose. If possible, I would also remove the sections on basement membranes and structural tissues and narrow the focus to the functions of GAGs in sensory tissues.

Author response

The suggested segments have all been removed from the revised manuscript as :-suggested to provide more focus, these deleted segments included :-

  1. Basement membranes as GAG interactive platforms
  2. The impact of HS deficiency
  3. Regulation of axonal development by GAGs,

6.1 Specific roles for lumican and keratocan in the formation of neural networks,

Table II Heparin/HS Interactive Proteins in the Cellular Environment Operative in Health and Disease,

6.1.1 The glypicans and syndecans as molecular targets in cancer therapy,

Table III Nuclear HS and HSPGs identified in a number of tumour cell types,

Table IV Nuclear HS and HSPGs identified in normal mammalian cell nuclei,

6.1.2 Viruses which bind to cell surface HS as part of the infective process,

6.1.3 HS endocytosis receptors,

Table V. Viruses that gain access to cells through interaction with cell surface HS, 6.2 Pathogenic bacteria interact with cell surface HS in an infective process,

Table VI. Pathogenic Bacteria Which Attach to Cell Surface HS as part of the Transmission of Infectious Diseases,

  1. HS Interactive proteins operative in health and disease,

7.1 The HS Interactome,

This represented 14 print pages, 3 figures and 4 Tables collectively ~40% of the manuscript. To replace this I undertook a comprehensive review of GAG and PG biodiversity and showed how this affects proteoglycans that provide a cell with the ability to detect its environment, as you also requested.

Reviewer comment

In addition, what’s the author’s definition of sensory tissue? There are extensive discussions of the role of proteoglycans in the synapse. Can cells in the CNS really be considered sensory tissue? I think it benefits the manuscript to focus on proteoglycans in tissues that sense the outside environment directly.

Author response

I take your point. Comments on sensory tissues as noted in your comments have been removed from the revised manuscript.

Author response

I have comprehensively covered proteoglycans that sense the outside environment in my revision and have pared down comments on the CNS/PNS and basement membrane but I feel the CNS examples that remain should still be part of the revision since they are important examples of how GAG biodiversity affects these proteoglycans in health and disease. After-all this is supposed to be the stated theme of this issue of IJMS.

Reviewer 2 Report

Comments and Suggestions for Authors

Review of manuscript ID: ijms-3442013 entitled ‘’ Glycosaminoglycan Components of Cell Associated and ECM 2 Proteoglycans Convey Molecular Recognition, and Homeostatic 3 Cell Instructive Capability to Structural and Sensory Tissues’

This article examines glycosaminoglycans (GAGs), a varied group of biomolecules that have developed over time as crucial elements in the glycocalyx surrounding all cells. When attached to cell surface and extracellular matrix (ECM) proteoglycans, GAGs possess molecular recognition and cell-instructive qualities that influence cellular behavior. The cell's ability to detect mechanical signals from changes in the ECM environment enables it to initiate appropriate biosynthetic responses to sustain ECM composition and tissue function. ECM proteoglycans with GAG attachments offer structural support to load-bearing tissues and enable them to resist shear forces in certain tissue contexts. This review highlights the structural intricacy of GAGs and the functional characteristics they impart to cellular and ECM proteoglycans. Proteoglycans play vital roles in cartilaginous load-bearing tissues and fibrocartilage subjected to tension and high shear forces. Certain proteoglycans also contribute to the regulation of neurophysiological responses in the central nervous system /PNS and provide instructive guidance in the formation of neural networks. This review provides examples of these roles. A more comprehensive understanding of the properties GAGs confer to tissues may aid in the development of GAG-based biotherapeutics designed to address specific instances of tissue dysfunction in disease processes and innovative tissue repair strategies.

Overall, the manuscript requires revisions before publication.

Specific Points:

  1. References are not used properly. In some instances, the authors used more than 6 References for one sentence whilst in some instances there are no References for scientific claims. Re-organize the References and make sure to provide References for any claim made.
  2. Long sentences- some sentences are more than 5 lines. Authors must shorten these sentences and be very clear of what they want to say. Be concise and straight to the point.
  3. Language- some sentences may require editing to make them scientifically sound. Do not use layman's language in a scientific manuscript.
  4. Introduction – this section is too short. A proper description of the ECM, GAGs and proteoglycans is required. Many reviews have been published and are available to the authors for Reference. Please see the following reviews on the ECM, GAGs and proteoglycans and write a minimum of 1,5 pages on the Introduction.

PMID: 38407002

PMID: 38500384

PMID: 35613357

PMID: 29499356

PMID: 32825245

PMID: 35268340

PMID: 37092398

PMID: 39334952

PMID: 23606640

PMID: 33605520

PMID: 21123617

PMID: 26562801

  1. Figures- make sure all Figures are clear for readers. Some parts of Figures – for example, Figure 1 are not clear. Readers must be able to see every part of an image. For example – some texts on Figure 9 are unreadable.
  2. This reviewer is assuming that Table 1 description is linked to Figure 3. Authors must clearly state this in Figure 3 or in Table 1 – that they are linked. Otherwise, Table 1 can be given as a sub-note of Figure 3.
  3. Sections of the manuscript must be properly formatted. Too long paragraphs – for example, section 2.4 must be divided into 2-3 paragraphs. Do this throughout the manuscript.
  4. What is the source of Figure 6C? Provide source. There is also an incomplete sentence on the Figure 6 legend. Correct this.
  5. Section 3.1- some text are given in italics. Why?

Comments on the Quality of English Language

Review of manuscript ID: ijms-3442013 entitled ‘’ Glycosaminoglycan Components of Cell Associated and ECM 2 Proteoglycans Convey Molecular Recognition, and Homeostatic 3 Cell Instructive Capability to Structural and Sensory Tissues’

This article examines glycosaminoglycans (GAGs), a varied group of biomolecules that have developed over time as crucial elements in the glycocalyx surrounding all cells. When attached to cell surface and extracellular matrix (ECM) proteoglycans, GAGs possess molecular recognition and cell-instructive qualities that influence cellular behavior. The cell's ability to detect mechanical signals from changes in the ECM environment enables it to initiate appropriate biosynthetic responses to sustain ECM composition and tissue function. ECM proteoglycans with GAG attachments offer structural support to load-bearing tissues and enable them to resist shear forces in certain tissue contexts. This review highlights the structural intricacy of GAGs and the functional characteristics they impart to cellular and ECM proteoglycans. Proteoglycans play vital roles in cartilaginous load-bearing tissues and fibrocartilage subjected to tension and high shear forces. Certain proteoglycans also contribute to the regulation of neurophysiological responses in the central nervous system /PNS and provide instructive guidance in the formation of neural networks. This review provides examples of these roles. A more comprehensive understanding of the properties GAGs confer to tissues may aid in the development of GAG-based biotherapeutics designed to address specific instances of tissue dysfunction in disease processes and innovative tissue repair strategies.

Overall, the manuscript requires revisions before publication.

Specific Points:

  1. References are not used properly. In some instances, the authors used more than 6 References for one sentence whilst in some instances there are no References for scientific claims. Re-organize the References and make sure to provide References for any claim made.
  2. Long sentences- some sentences are more than 5 lines. Authors must shorten these sentences and be very clear of what they want to say. Be concise and straight to the point.
  3. Language- some sentences may require editing to make them scientifically sound. Do not use layman's language in a scientific manuscript.
  4. Introduction – this section is too short. A proper description of the ECM, GAGs and proteoglycans is required. Many reviews have been published and are available to the authors for Reference. Please see the following reviews on the ECM, GAGs and proteoglycans and write a minimum of 1,5 pages on the Introduction.

PMID: 38407002

PMID: 38500384

PMID: 35613357

PMID: 29499356

PMID: 32825245

PMID: 35268340

PMID: 37092398

PMID: 39334952

PMID: 23606640

PMID: 33605520

PMID: 21123617

PMID: 26562801

  1. Figures- make sure all Figures are clear for readers. Some parts of Figures – for example, Figure 1 are not clear. Readers must be able to see every part of an image. For example – some texts on Figure 9 are unreadable.
  2. This reviewer is assuming that Table 1 description is linked to Figure 3. Authors must clearly state this in Figure 3 or in Table 1 – that they are linked. Otherwise, Table 1 can be given as a sub-note of Figure 3.
  3. Sections of the manuscript must be properly formatted. Too long paragraphs – for example, section 2.4 must be divided into 2-3 paragraphs. Do this throughout the manuscript.
  4. What is the source of Figure 6C? Provide source. There is also an incomplete sentence on the Figure 6 legend. Correct this.
  5. Section 3.1- some text are given in italics. Why?

Author Response

Manuscript ID: ijms-3442013 entitled ‘’ Glycosaminoglycan Components of Cell Associated and ECM Proteoglycans Convey Molecular Recognition, and Homeostatic Cell Instructive Capability to Structural and Sensory Tissues

Please note this manuscript has a new title. New Title: “Glycosaminoglycans, Instructive Biomolecules That Regulate Cellular Activity And Synaptic Neuronal Control Of Specific Tissue Functional Properties”.

Point by point responses to reviewer 2.

Reviewer 2

Reviewer comments: Specific Points:

References are not used properly. In some instances, the authors used more than 6 References for one sentence whilst in some instances there are no References for scientific claims. Re-organize the References and make sure to provide References for any claim made.

Author response

The reference citation usage has been modified in the revision taking your points on board.

Reviewer comment

Long sentences- some sentences are more than 5 lines. Authors must shorten these sentences and be very clear of what they want to say. Be concise and straight to the point.

Author response

The sentences have been re-structured as you have suggested.

Reviewer comment

Language- some sentences may require editing to make them scientifically sound. Do not use layman's language in a scientific manuscript.

Author response

Various segments of the manuscript have been re-structured as you have suggested.

Reviewer comment

Introduction – this section is too short. A proper description of the ECM, GAGs and proteoglycans is required. Many reviews have been published and are available to the authors for Reference. Please see the following reviews on the ECM, GAGs and proteoglycans and write a minimum of 1.5 pages on the Introduction.

Author response

Most of the suggested material has been incorporated into the amended introduction section in the revision which is now expanded to the length suggested.

Reviewer comment

Figures- make sure all Figures are clear for readers. Some parts of Figures – for example, Figure 1 are not clear. Readers must be able to see every part of an image. For example – some texts on Figure 9 are unreadable.

Author response

All figures have been revised in the revised manuscript taking your point on board.

Reviewer comment

This reviewer is assuming that Table 1 description is linked to Figure 3. Authors must clearly state this in Figure 3 or in Table 1 – that they are linked. Otherwise, Table 1 can be given as a sub-note of Figure 3.

Author response

This figure has been re-organised to make this clearer in the revised manuscript.

Reviewer comment

Sections of the manuscript must be properly formatted. Too long paragraphs – for example, section 2.4 must be divided into 2-3 paragraphs. Do this throughout the manuscript.

Author response

The revised manuscript has been duly reformatted as requested.

Reviewer comment

What is the source of Figure 6C? Provide source. There is also an incomplete sentence on the Figure 6 legend. Correct this.

Author response

Source of figure now provided and sentence corrected.

Reviewer comment

Section 3.1- some text are given in italics. Why?

Author response

This is a puzzle to me, no italics segments were in my version of the manuscript which I submitted and I have checked they do not appear in the revised version of the manuscript. I can only presume that these were some glitch that appeared on electronic transfer.

Round 2

Reviewer 2 Report

Comments and Suggestions for Authors

The author addressed all my concerns